# INVERSION-GUIDED WEIGHT MASKING AND PATCHING: PURIFICATION-FREE DEFENSE FOR DIFFUSION MODELS AGAINST ADVERSARIAL PERTURBATION

## ABSTRACT

Diffusion models have demonstrated outstanding generative capabilities but remain vulnerable to adversarial perturbations. These perturbations, originally intended for data copyright protection, expose a critical robustness weakness in diffusion models. Existing defenses mainly based on data purification, which require prior assumptions and incur high computational costs. In this work, we investigate the impact of perturbations on different modules of diffusion models and introduce Inversion-Guided Weight Masking and Patching (IMAP), a purification-free method designed to restore diffusion models customized on adversarially perturbed data. Our approach first applies a prompt re-mapping strategy before customization. We then use DDIM inversion to identify critical convolutional kernels affected by perturbations and perform weight masking and adaptive patching to restore the model. IMAP requires no clean data or costly per-image purification. Extensive experiments on CelebA-HQ and VGGFace2 demonstrate that IMAP significantly improves generation quality under various adversarial scenarios. Furthermore, through comparisons with purification-based techniques, we demonstrate the effectiveness of IMAP and show that it can be effectively integrated with existing purification-based methods.

## 1 INTRODUCTION

Diffusion models have achieved remarkable success across various generative tasks (Ho et al., 2020; Song et al., 2021b; Rombach et al., 2022), including text-to-image synthesis (Song et al., 2021b; Rombach et al., 2022; Dhariwal & Nichol, 2021; Gafni et al., 2022), image-to-image translation (Saharia et al., 2022; Parmar et al., 2023), image editing (Meng et al., 2022; Hertz et al., 2023; Tumanyan et al., 2023), and time-series applications (Croitoru et al., 2023; Lin et al., 2024). Among these, Stable Diffusion (Rombach et al., 2022) stands out for its strong generative ability and flexibility. Moreover, existing methods like DreamBooth (Ruiz et al., 2023) enable efficient customization using only a small amount of samples, and boost the wide adoption of diffusion models in generative applications.

Despite their generative ability, diffusion models exhibit a critical weakness: low robustness to adversarial attacks (Van Le et al., 2023; Liang et al., 2023; Xue et al., 2023; Shan et al., 2023; Liang & Wu, 2023; Liu et al., 2024b). In particular, customized models fine-tuned on adversarial data can suffer from severe degradation in synthesis quality. Data providers can add carefully designed perturbations to the clean data to generate adversarial data, especially in text-to-image scenario. These perturbations preserve semantic meaning but severely mislead the model during learning. Specifically, when given a text prompt corresponding to a target concept such as a specific identity or painting style, the model fails to generate accurate and high-quality images that match the target concept. However, previous studies have also observed that when generating images with prompts different from those used during training, customized diffusion models can still reproduce the targeted concept with high quality (Liu et al., 2024a; Wan et al., 2024). This indicates that the model has actually learned the perturbed concept. Liu et al. further point out that data perturbations exploit the shortcut learning vulnerabilities of customized diffusion models, causing a latent-space misalignment between images and prompts (Liu et al., 2024a).

Figure 1: **Illustration of shortcut learning issue and our approach.** Top row shows the effect of adversarial perturbation on diffusion model customization, and the rest rows illustrate our approach.

To address such perturbations, recent work has focused mainly on purification-based methods (Zhao et al., 2024; Cao et al., 2023; Hönig et al., 2024). These approaches aim to remove the perturbation from the data before customization by some image purification techniques. While effective to some extent, purification-based methods face several limitations. First, without prior knowledge of the clean data distribution, purification can introduce uncontrolled alterations to critical content. The model's ability to learn the intended concept then depends on whether the purified data match the clean distribution. Second, these methods often require expensive, per-image optimization steps.

In this work, we investigate the impact of adversarial perturbations on different modules of diffusion models and propose Inversion-Guided Weight Masking and Patching (IMAP): a purification-free restoration for diffusion models on perturbed data. We address shortcut vulnerabilities at the model level by restoring compromised shortcuts, as shown in Figure 1. Our approach first applies a prompt re-mapping technique before customization. We then leverage DDIM inversion (Song et al., 2021a) to investigate the convolutional layers that highly correlate with perturbations. DDIM inversion allows us to reverse the diffusion process and reconstruct the initial noise from a generated image. To our best knowledge, this is the first application of DDIM inversion for defending diffusion models against data perturbations. Finally, we perform weight masking and adaptive patching to restore the model. Extensive experiments under various perturbations demonstrate that IMAP-defended models better generate the intended concepts. In summary, our contributions are as follows:

- We investigate how data perturbations impact diffusion models and propose a novel method, IMAP, to handle these challenging scenarios. IMAP restores abnormal shortcuts, effectively eliminating generation inconsistencies without needing access to clean data or relying on data purification.

- We conduct extensive experiments on CelebA-HQ (Karras et al., 2018) and VGGFace2 (Cao et al., 2018), comparing various perturbation methods using multiple metrics. The results demonstrate the effectiveness of our approach.

- We further conduct comparative experiments with two advanced purification-based methods (Hönig et al., 2024; Cao et al., 2023), and the experimental results demonstrate the superiority of our approach. Additionally, our method can be integrated with these purification-based techniques to further enhance performance.

## 2 RELATED WORKS

**Customized text-to-image diffusion models.** Text-to-image diffusion models (Ho et al., 2020; Song et al., 2021b; Rombach et al., 2022; Dhariwal & Nichol, 2021; Gafni et al., 2022) generate

high-fidelity images from textual prompts using U-Net (Ronneberger et al., 2015) based denoising backbones. Latent diffusion models (LDMs) (Rombach et al., 2022) further improve efficiency by operating in a learned latent space. To condition image generation, prompts are embedded via text encoders and injected into the diffusion process. Recent methods such as DreamBooth (Ruiz et al., 2023), Textual Inversion (Gal et al., 2023), and ControlNet (Zhang et al., 2023) allow for low-cost customization on small-scale data.

**Adversarial perturbations and defenses in diffusion models.** Recent works have introduced data perturbations into diffusion pipelines to protect data from unauthorized usage (Van Le et al., 2023; Liang et al., 2023; Xue et al., 2023; Shan et al., 2023; Salman et al., 2023). These perturbations, inspired by adversarial attacks on classification models, aim to preserve semantic meaning in data for human recognition yet harmful to generative models. PhotoGuard (Salman et al., 2023) counters image editing. Glaze (Shan et al., 2023) prevents diffusion models from mimicking an artist's style. Anti-DreamBooth (Van Le et al., 2023) targets DreamBooth customization method. AdvDM and SDS serires (Liang et al., 2023; Xue et al., 2023) optimize perturbations to interfere with learning, while PAP (Wan et al., 2024) proposes the first prompt-agnostic method.

Defenses include purification-based approaches such as IMPRESS (Cao et al., 2023), GrIDPure (Zhao et al., 2024), and Noisy-Upscaling (Hönig et al., 2024), but they incur high overhead and risk distorting useful content. Liu et al. (2024a) propose a causal defense via contrastive decoupling, still requiring data purification. Model unlearning has also been explored as a defense against adversarial attacks (Truong et al., 2025), but most approaches rely on clean data for concept restoration, making them unsuitable for scenarios where such data is unavailable. To the best of our knowledge, existing diffusion-related works lack methods that address adversarial data by leveraging the model's own capacity instead of external priors.

**Shortcut learning.** Shortcut learning occurs when models rely on spurious correlations instead of core semantics (Geirhos et al., 2020). This phenomenon underlies both backdoor attacks (Wang et al., 2019) and adversarial vulnerabilities. Liu et al. (2024a) highlight how adversarial examples in customized diffusion models can exploit such shortcuts. While this issue is well-studied in classification tasks, shortcut learning in diffusion models remains relatively underexplored.

## 3 PRELIMINARIES

**Diffusion process.** Diffusion models are a type of generative models, including a forward diffusion process and a reverse process. Let $\epsilon \sim N(0, 1)$ and $\bar{\alpha}_t$ denotes $\prod_{i=1}^{t} \alpha_i$. For a given data sample $x_0$ and timestep $t \in [1, T]$, diffusion model gradually adds Gaussian noise onto it to get $x_t$ according to Eq. (1) in the forward process, where $\alpha_t = 1 - \beta_t$ and $\beta_t \in (0, 1)$ is the variance schedule.

$$x_t = \sqrt{\alpha_t} x_{t-1} + \sqrt{1 - \alpha_t} \epsilon = \sqrt{\bar{\alpha}_t} x_0 + \sqrt{1 - \bar{\alpha}_t} \epsilon_t, \tag{1}$$

In the reverse process, the model learns to denoise from $x_{t+1}$ by minimizing the $\mathscr{L}_2$ distance between the noise $\hat{\epsilon}_t$ predicted by the neural network $\theta$ and the true noise $\epsilon_t$ used in the forward process. The loss function for conditional diffusion models with text prompt $c$ is as follows:

$$\mathcal{L}_{cond}^{DM}(\theta, x_0) = \mathbb{E}_{x_0, t, c, \epsilon \in N(0,1)} ||\epsilon - \hat{\epsilon}_{x_{t+1}, t, c}||_2^2. \tag{2}$$

**DreamBooth.** DreamBooth (Ruiz et al., 2023) is a type of parameter efficient fine-tuning method to personalize text-to-image diffusion models. For a given concept '*sks*' and the class name '[class noun]' of the concept, DreamBooth learns the new concept by a generic prompt $c$, such as "a photo of *sks* [class noun]", and a prior prompt $c_{pr}$ like "a photo of [class noun]". First, DreamBooth generates a set of class images randomly with the frozen pretrained model and $c_{pr}$. To fine-tune the pretrained model on concept *sks* while preventing *language drift*, based on Eq. (2), DreamBooth employs a two-part training loss as Eq. (3):

$$\mathcal{L}_{db}^{DM}(\theta, x_0) = \mathbb{E}_{x_0, t, t'} ||\epsilon - \hat{\epsilon}_{x_{t+1}, t, c}||_2^2 + \lambda ||\epsilon_{pr} - \hat{\epsilon}_{x'_{t'+1}, t', c_{pr}}||_2^2, \tag{3}$$

where $\epsilon, \epsilon_{pr} \in N(0, 1)$, $\lambda$ is a hyperparameter that adjusts the importance of prior loss, $x'$ is sampled from the class-image dataset, and $x'_{t'+1}$ is the noisy variable of $x'$.

**DDIM inversion.** Given an image $x_0$, using the deterministic DDIM (Song et al., 2021a) formulation, DDIM inversion estimates the noise latent $x_T$ such that the denoising process approximately reconstructs $x_0$ under a fixed noise schedule by approximating $\hat{\epsilon}_{x_{t+1},t,c}$ with $\hat{\epsilon}_{x_t,t,c}$:

$$x_{t+1} = \frac{\sqrt{\alpha_{t+1}}}{\sqrt{\alpha_t}}\left(x_t - \sqrt{1-\alpha_t}\cdot\hat{\epsilon}_{x_t,t,c}\right) + \sqrt{1-\alpha_{t+1}}\cdot\hat{\epsilon}_{x_t,t,c}. \tag{4}$$

## 4 METHODOLOGY

### 4.1 MOTIVATION

Building on the observation by Liu et al. (2024a) that adversarial perturbations cause shortcut learning by creating a mismatch between images and prompts, we conducted controlled experiments to isolate their effects on the text encoder (TE) and U-Net. Our detailed analysis, provided in Appendix A, reveals that each component plays a distinct role in generating perturbed outputs. We found that a perturbed TE embeds misleading, noise-like semantics, leading to artifacts like distorted backgrounds. A perturbed U-Net then reinforces this behavior by adopting faster but less stable generation paths. A further investigation localized the most significant adversarial effects to specific U-Net modules, particularly UpBlock1 and DownBlock2/3. Motivated by these findings, our work proposes a comprehensive solution to repair the vulnerabilities in both the U-Net and the TE, allowing us to fix the model with minimal intervention.

### 4.2 IMAP: INVERSION-GUIDED WEIGHT MASKING AND PATCHING

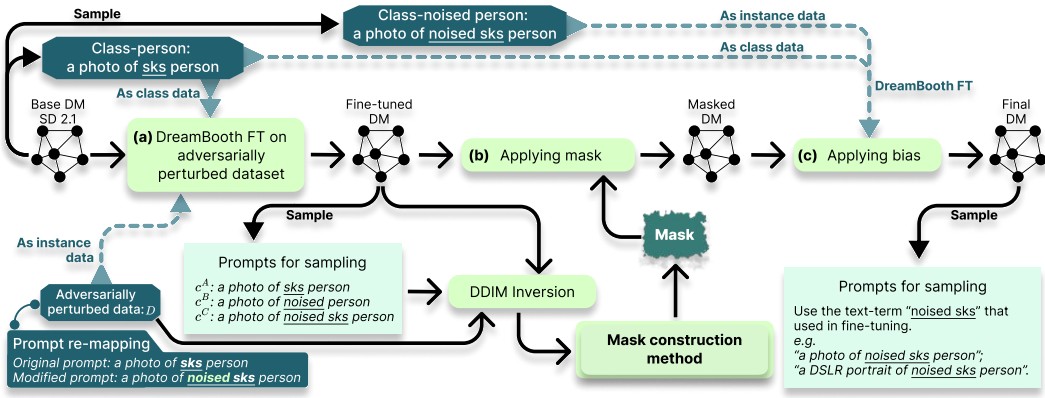

Figure 2: **Method Overview.** Our method consists of three stages: **(a)** fine-tuning, **(b)** weight masking, and **(c)** patching. In **(a)**, we re-map the text prompt before DreamBooth fine-tuning to separate concept and perturbation. In **(b)**, we apply a weight mask to the fine-tuned model, guided by DDIM inversion. In **(c)**, we perform adaptive patching using data synthesized by the base model.

In this section, we study a challenging scenario where a diffusion model is fine-tuned solely on perturbed data without access to the corresponding clean images. These perturbations are visually imperceptible but are intentionally crafted to manipulate the model's behavior at inference time. Our goal is to understand how such perturbations affect the internal components of the model, and to recover faithful generation performance despite the absence of clean supervision.

Let $D = \{(x^i, c)\}_{i=1}^N$ be a dataset of perturbed images $x^i = f_p(x_{clean}^i)$, where $f_p$ is an unknown data perturbation function, and $c$ is the shared training prompt. Let $\theta_{pre}$ denote the parameters of a pretrained diffusion model, and $\theta$ the parameters after DreamBooth fine-tuning on dataset $D$. Due to the perturbations, the model $\theta$ tends to generate anomalous or irrelevant content, even when provided with a correct prompt. Our goal is to estimate updated model parameters $\theta^*$, such that:

$$\forall c \in \mathcal{C}, \quad \mathcal{G}(\theta^*, c) \approx \mathcal{G}(\theta_{clean}, c), \tag{5}$$

where $\mathcal{G}(\theta, c)$ denotes the image generated by model $\theta$ given prompt $c$, and $\theta_{clean}$ is a hypothetical model fine-tuned on the clean dataset $\{(x_{clean}^i, c)\}$, which is not accessible in practice.

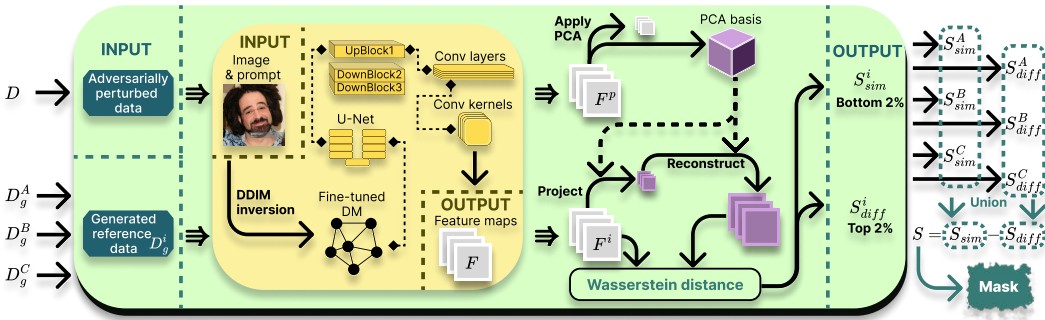

Figure 3: **Mask construction method.** The method output a mask for given $D$ and $D_g^i$ with a preset threshold (2% in the figure), where $i \in \{A, B, C\}$. In DDIM inversion process, we focus on the convolutional layers of the target blocks as we mentioned in Section 4.1, and we obtain the feature maps to distinct kernels that contribute most in shortcut generating.

To address this challenge, we propose a lightweight correction framework with three stages: fine-tuning, weight masking, and patching. We first fine-tune the model using only perturbed samples via DreamBooth, with slightly modified prompts to semantically decouple the protected content from the perturbation. In the masking stage, we analyze intermediate U-Net activations to identify a small set of kernels consistently correlated with adversarial signals and set them to zero. Finally, we patch the model using synthetic data generated by the pretrained model to restore the target concept without reinforcing adversarial patterns. An overview of the pipeline is shown in Figure 2.

**Prompt re-mapping for concept-perturbation separation.** In our fine-tuning stage, we introduce a *prompt re-mapping* strategy to link the perturbed concept to a specific term rather than the original one. For a concept '*sks*' and its class name '[class noun]', we use the prior prompt $c_{pr}$ like "a photo of [class noun]" for DreamBooth. Instead of the standard instance prompt "a photo of sks [class noun]", we modify it to "a photo of noised sks [class noun]". This adjustment explicitly conditions the model to associate the perturbed concept with the term 'noised sks' instead of 'sks', allowing us to leverage the differences in the generation process between terms–'sks', 'noised', and 'noised sks'–and the perturbed images. It helps isolate perturbation-related activations, enabling targeted masking that suppresses spurious kernels while preserving core concept features.

**Inversion-guided weight masking.** Liu et al. (2024a) show that adversarial perturbations trigger shortcut learning in customized diffusion models. Neural Cleanse (Wang et al., 2019) attributes the success of backdoor attacks in DNNs to shortcut learning, and mitigates them via trigger inversion, masking, and patching. Inspired by this, we analyze generation in DMs via DDIM inversion, apply masking to suppress perturbation-sensitive activations, and patch with clean data to recover fidelity. See Figure 3.

In our masking stage, after fine-tuning the model on perturbed dataset $D$, we synthesize three reference datasets $D_g = \{D_g^A, D_g^B, D_g^C\}$ using prompts: $c^A$–"a photo of sks [class noun]", $c^B$–"a photo of noised [class noun]" and $c^C$–"a photo of noised sks [class noun]". For each $\{D_g^i, c^i\}$, we apply DDIM inversion to extract feature maps $F^i$ of the convolutional layers. Similarly, we obtain $F^p$ for the perturbed training set $D$. We compute distances $Dist(F^i, F^p)$ and select kernels with high similarity or difference under a predefined threshold $Thr$:

$$S_{sim}^i, S_{diff}^i = Select(Dist(F^i, F^p), Thr), \quad S_{sim} = \bigcup S_{sim}^i, \quad S_{diff} = \bigcup S_{diff}^i. \quad (6)$$

Inspired of Plug-and-Play (Tumanyan et al., 2023) and FreeControl (Mo et al., 2024), we build semantic basis for $F^p$ via PCA to handle the high dimension vectors. We then compute the Wasserstein distance (WD) between $F^i$ and its PCA-based reconstruction:

$$basis, F_{PCA}^p = PCA(F^p), Dist(F^i, F^p) = WD(F^i, Recon(Proj(F^i, basis))). \quad (7)$$

Empirically, $S_{sim}$ tends to capture low-quality artifacts shared across the three prompts due to perturbation-induced shortcuts. In contrast, $S_{diff}$ highlights kernels sensitive to semantic changes. Thus, we define the final mask as $S = S_{sim} - S_{diff}$, preserving concept-relevant activations while suppressing perturbation-sensitive ones.

**Adaptive patching via targeted fine-tuning.** Even though the masking stage zeroes out kernels that are highly correlated with the perturbations, the model still fails to generate ideal images. From the perspective of shortcut learning, this is because, despite removing the most perturbation-sensitive kernels, the specified prompt can still induce the remaining kernels to shift the generation process toward the shortcut. Similar findings have been observed in DNNs: NC demonstrated that while the top 1% of neurons are highly correlated with backdoor-induced shortcuts, removing at least 30% of the neurons is necessary to completely eliminate the effect (Wang et al., 2019). This phenomenon can be attributed to the massive redundancy in neural pathways in DNNs (Hu et al., 2016).

However, in diffusion models, we cannot afford to zero out numerous kernels, as this would significantly reduce the model's feature extraction capability, leading to the inability to generate meaningful images. Thus, we further introduce an *adaptive patching* strategy based on our earlier *prompt re-mapping*. It repairs masked U-Net kernels and restores the text encoder's capacity to capture task-relevant semantics, effectively suppressing shortcuts without compromising content fidelity.

In the previous stages, we fine-tuned the model using the modified prompt $c^C$, linking the perturbed concept to the term 'noised sks'. In *adaptive patching* stage, our goal is to restore the guiding role of the term 'noised'. Specifically, we leverage the pretrained model (i.e., the base model before fine-tuning) to generate a dataset $D_{pt}$ using prompt $c^B$ with 'noised' term in advance. We then fine-tune the pruned model with $D_{pt}$ via DreamBooth to perform adaptive model-parameter correction, ensuring that both the U-Net and text encoder are restored and the term 'noised' can correctly contribute to the generation process. In this stage, we use the prior class prompt $c_{pr}$ as before. This patching step serves to fix the pruned kernels by reinforcing the model's ability to handle the 'noised' text term while preventing it from falling back to shortcut-based generation.

## 5 EXPERIMENTS

### 5.1 EXPERIMENTAL SETUP

**Datasets and perturbations.** We conduct our experiments on human-face generation task using the CelebA-HQ (Karras et al., 2018) and VGGFace2 (Cao et al., 2018) datasets. For each dataset, we randomly choose 5 different identites, and 4 clean images per identity with adversarial images conducted via seven types of perturbations: AntiDB, AdvDM(+/-), Glaze, SDS(+/-), SDST, where (+/-) denotes the use of gradient ascent/descent to construct the perturbation.

**Customization settings.** We choose the latest Stable Diffusion v2.1 as the pretrained base DM, and adopt DreamBooth (Ruiz et al., 2023) to customize DMs on both clean and adversarial data. We generate 200 pics by the base DM to build the class images for DreamBooth using the prior prompt $c_{pr}$ : "a photo of person". For each model, we use a text prompt "a photo of [term] person"as instance prompt $c$ for training and two prompts: $c_{train}$-"a photo of [term] person" and $c_{diff}$-"a dslr portrait of [term] person" for sampling, where [term] is settled as 'sks' for the clean fine-tuned models and 'noised sks' for both baseline our IMAP restored models under different perturbations. We run experiments on 4 NVIDIA RTX A6000 (48G) GPUs. More details are provided in Appendix B.2.

**Evaluation metrics.** We evaluate our IMAP restored method by generating 30 pics per prompt per model. The Full Reference (FR) metrics, FID (Heusel et al., 2017), SSIM (Wang et al., 2004), and PSNR compare images generated by our baseline or IMAP models against those generated by the clean-ft models in terms of distributional similarity, structural fidelity, and pixel-level accuracy, and the No Reference (NR) metrics, TOPIQ (topiq_nr_swin_face) (Chen et al., 2024; Qin et al., 2023) and QAlign (qalign_8bit) (Wu et al., 2024) measure face quality without reference. All metrics are implemented using the IQA-PyTorch toolbox (Chen & Mo, 2022).

### 5.2 EFFECTIVENESS OF IMAP UNDER DIFFERENT ADVERSARIAL PERTURBATIONS

Our IMAP method aims to mitigate the impact of adversarial perturbations on customized DMs, while preserving the integrity of the newly learned concept. We evaluate its effectiveness through a series of comparative experiments, where different perturbation strategies are applied.

Figure 4 illustrates representative image generation results under various adversarial perturbations on the VGGFace2 dataset. As shown in the figure, our IMAP defense significantly improves gener-

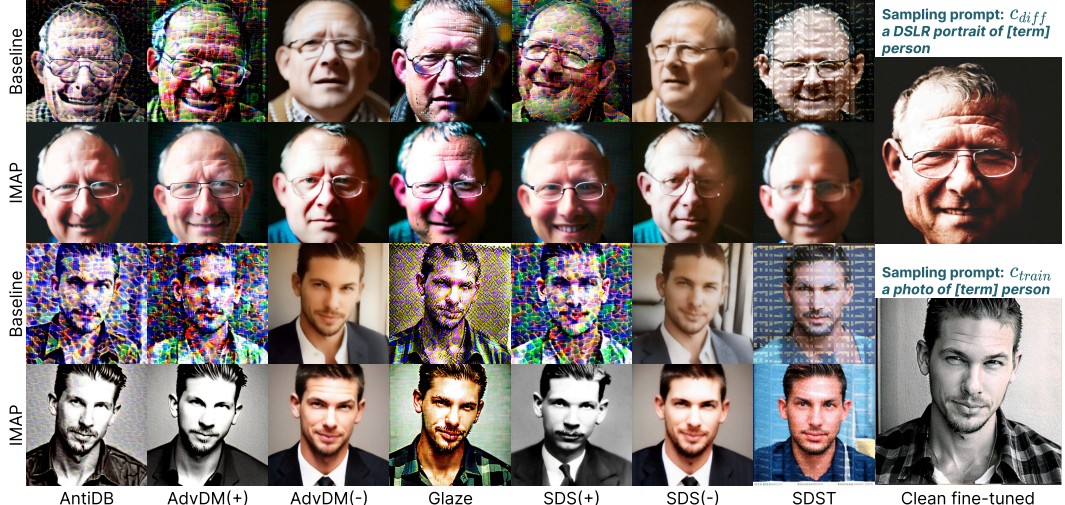

Figure 4: **Visual comparison of images generated by customized diffusion models under different adversarial perturbations(with and without IMAP).** Each column corresponds to a different perturbation method applied during fine-tuning. IMAP consistently improves robustness against perturbations while preserving the identity of the learned concept. See Appendix for more results.

Table 1: **Quantitative evaluation on VGGFace2 across multiple metrics under different sampling prompt settings.** ↓/↑ indicate that lower/higher values are better. We compare IMAP-defensed models against baselines customized on various types of perturbed data. Results are reported under both sampling prompts ($c_{train}$ and $c_{diff}$) as well as individually. Each value is the average metric over models trained on 5 data identities per dataset; see Appendix for results under each identity.

| Adversarial Perturbations | FID↓ | | PSNR↑ | | SSIM↑ | | TOPIQ↑ | | QAlign↑ | |
|---|---|---|---|---|---|---|---|---|---|---|
| | Baseline | IMAP | Baseline | IMAP | Baseline | IMAP | Baseline | IMAP | Baseline | IMAP |
| *Both prompts: $c_{train}, c_{diff}$* | | | | | | | | | | |
| AntiDB | 377.84 | **159.41** | 9.38 | **10.96** | 0.21 | **0.35** | 0.20 | **0.56** | 1.55 | **2.75** |
| AdvDM(+) | 397.33 | **184.83** | 9.05 | **10.76** | 0.20 | **0.34** | 0.13 | **0.54** | 1.43 | **2.48** |
| AdvDM(-) | 147.99 | **136.43** | 11.12 | **11.72** | 0.38 | **0.42** | 0.34 | **0.46** | 1.97 | **2.59** |
| Glaze | 224.52 | **170.91** | 9.11 | **10.39** | 0.21 | **0.31** | 0.34 | **0.51** | 1.60 | **2.59** |
| SDS(+) | 378.86 | **183.22** | 9.38 | **10.86** | 0.21 | **0.35** | 0.16 | **0.55** | 1.49 | **2.81** |
| SDS(-) | 147.67 | **142.90** | 11.10 | **11.78** | 0.38 | **0.42** | 0.32 | **0.39** | 1.91 | **2.31** |
| SDST | 327.41 | **199.61** | 10.02 | **10.92** | 0.28 | **0.36** | 0.35 | **0.47** | 2.12 | **2.54** |
| *Train prompt: $c_{train}$* | | | | | | | | | | |
| AntiDB | 438.98 | **206.38** | 8.84 | **9.67** | 0.18 | **0.27** | 0.12 | **0.47** | 1.51 | **2.15** |
| AdvDM(+) | 453.04 | **253.01** | 8.59 | **9.45** | 0.18 | **0.25** | 0.07 | **0.42** | 1.42 | **1.72** |
| AdvDM(-) | **167.60** | 186.53 | **10.92** | 10.47 | 0.35 | 0.35 | 0.35 | **0.38** | 2.04 | **2.23** |
| Glaze | 234.71 | **224.88** | 8.58 | **8.83** | 0.16 | **0.19** | 0.30 | **0.33** | 1.39 | **1.65** |
| SDS(+) | 428.63 | **258.50** | 8.85 | **9.58** | 0.19 | **0.25** | 0.12 | **0.40** | 1.43 | **2.02** |
| SDS(-) | **172.89** | 187.84 | **10.68** | 10.67 | 0.34 | **0.36** | 0.33 | **0.34** | 1.94 | **2.10** |
| SDST | 332.71 | **269.40** | 10.00 | **10.05** | 0.26 | **0.29** | **0.39** | 0.38 | **2.25** | 2.08 |
| *Diff prompt: $c_{diff}$* | | | | | | | | | | |
| AntiDB | 361.33 | **165.05** | 9.92 | **12.24** | 0.24 | **0.44** | 0.27 | **0.66** | 1.60 | **3.35** |
| AdvDM(+) | 380.05 | **166.96** | 9.50 | **12.07** | 0.23 | **0.43** | 0.18 | **0.66** | 1.45 | **3.24** |
| AdvDM(-) | 174.16 | **131.46** | 11.32 | **12.97** | 0.42 | **0.50** | 0.34 | **0.53** | 1.90 | **2.95** |
| Glaze | 257.25 | **164.92** | 9.64 | **11.95** | 0.25 | **0.42** | 0.39 | **0.70** | 1.81 | **3.52** |
| SDS(+) | 367.77 | **160.44** | 9.92 | **12.15** | 0.23 | **0.44** | 0.20 | **0.69** | 1.55 | **3.60** |
| SDS(-) | 162.26 | **140.05** | 11.53 | **12.90** | 0.42 | **0.49** | 0.32 | **0.45** | 1.87 | **2.51** |
| SDST | 366.41 | **178.62** | 10.04 | **11.78** | 0.30 | **0.44** | 0.32 | **0.56** | 1.98 | **3.01** |

Table 2: **Quantitative evaluation on CelebA-HQ across multiple metrics using both sampling prompts.** See Appendix for results under each prompt individually.

| Adversarial Perturbations | FID↓ | | PSNR↑ | | SSIM↑ | | TOPIQ↑ | | QAlign↑ | |
| --- | --- | --- | --- | --- | --- | --- | --- | --- | --- | --- |
| | Baseline | IMAP | Baseline | IMAP | Baseline | IMAP | Baseline | IMAP | Baseline | IMAP |
| AntiDB | 226.25 | **137.58** | 10.29 | **11.08** | 0.22 | **0.34** | 0.48 | **0.64** | 2.12 | **3.08** |
| AdvDM(+) | 387.31 | **182.72** | 9.56 | **10.87** | 0.21 | **0.34** | 0.12 | **0.53** | 1.46 | **2.51** |
| AdvDM(-) | 137.77 | **134.69** | 11.75 | **12.09** | 0.38 | **0.42** | 0.37 | **0.47** | 2.09 | **2.63** |
| Glaze | 215.03 | **171.86** | 9.89 | **10.69** | 0.20 | **0.29** | 0.47 | **0.54** | 2.23 | **2.78** |
| SDS(+) | 367.32 | **162.34** | 9.78 | **10.95** | 0.21 | **0.34** | 0.21 | **0.56** | 1.62 | **2.81** |
| SDS(-) | **132.55** | 133.95 | 11.87 | **12.18** | 0.39 | **0.42** | 0.35 | **0.44** | 2.01 | **2.46** |
| SDST | 268.10 | **164.48** | 10.21 | **10.97** | 0.25 | **0.37** | 0.40 | **0.47** | **2.58** | 2.42 |

ation quality compared to the baseline, especially under AntiDB, AdvDM(+) and SDS(+) perturbations. In these settings, the baseline models fail to generate recognizable human faces and instead produce outputs heavily corrupted with structured noise patterns. In contrast, IMAP effectively mitigates or even eliminates such artifacts, yielding cleaner and more realistic results.

For AdvDM(-) and SDS(-), while baseline outputs are blurry yet still depict identifiable faces, IMAP generates sharper and more coherent results. Under Glaze and SDST, IMAP also brings noticeable improvements, though to a lesser extent. In particular, SDST introduces structured noise that IMAP cannot fully remove but significantly mitigates, leading to higher-fidelity face reconstructions.

Moreover, IMAP suppresses noise more effectively when sampling with the fine-tuning prompt $c_{\text{train}}$ used during fine-tuning. With a different prompt $c_{\text{diff}}$, it continues to generate sharp images while better preserving key facial attributes, demonstrating its generalization beyond the training prompt.

Table 1 presents quantitative results on the VGGFace2 dataset, comparing IMAP against different baselines under three prompt settings: both prompts ($c_{\text{train}}$ and $c_{\text{diff}}$) jointly, and each individually. Under the joint setting, IMAP significantly improves generation performance across all metrics. Notably, it reduces FID by up to 58% (e.g., AntiDB: 377.84 → 159.41) achieves a threefold improvement in TOPIQ (e.g., AntiDB, AdvDM(+), SDS(+)), demonstrating strong robustness and effectiveness. Overall, IMAP consistently yields greater improvements under $c_{\text{diff}}$ than $c_{\text{train}}$. For instance, in the $c_{\text{train}}$ setting with AdvDM(-) and SDS(-), IMAP slightly underperforms the baseline in FID and PSNR, while still outperforming it in other metrics. This may be because these perturbations primarily cause mild blurring rather than major distributional shifts. Moreover, since $c_{\text{train}}$ is used during fine-tuning, it may be more tightly entangled with the perturbation, making it harder to correct distributional shifts than to improve perceptual facial quality. Results on CelebA-HQ dataset under both prompts are reported in Table 2, with per-prompt results included in Appendix D.2.

## 5.3 COMPARATIVE EXPERIMENTS BETWEEN IMAP AND DIFFERENT PURIFICATION METHODS

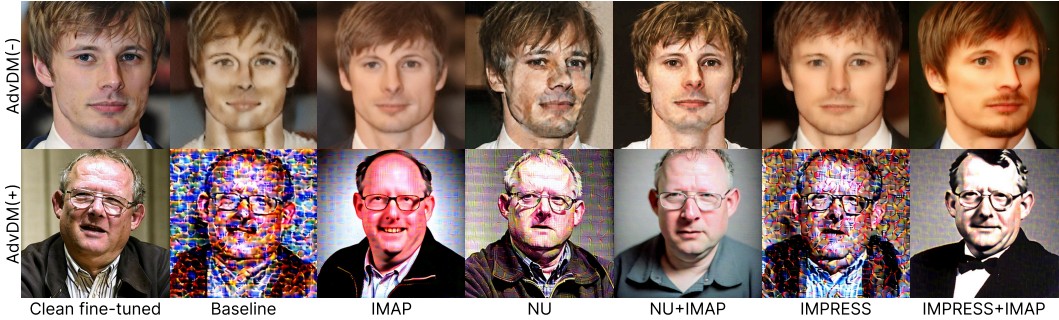

Figure 5: **Visual comparison results between IMAP and different purification-based methods.**

To validate the efficiency and practical applicability of IMAP, we selected two advanced purification methods for comparison: Noisy-Upscaling (Hönig et al., 2024) and IMPRESS (Cao et al., 2023).

Table 3: **Quantitative evaluation on VGGFace2 across multiple metrics under two experimental conditions: using only the purification method versus using both purification and IMAP.** See Appendix for results under each prompt individually.

| Adversarial Perturbations | FID↓ | | PSNR↑ | | SSIM↑ | | TOPIQ↑ | | Qalign↑ | |
|---|---|---|---|---|---|---|---|---|---|---|
| | Purify only | Purify +IMAP | Purify only | Purify +IMAP | Purify only | Purify +IMAP | Purify only | Purify +IMAP | Purify only | Purify +IMAP |
| *Purification method:Noisy-Upscaling* | | | | | | | | | | |
| AntiDB | **107.58** | 116.45 | 11.09 | **11.19** | 0.38 | **0.39** | 0.60 | **0.62** | 3.10 | **3.19** |
| AdvDM(+) | 204.58 | **143.68** | 9.95 | **10.80** | 0.25 | **0.34** | 0.38 | **0.55** | 1.75 | **2.71** |
| AdvDM(-) | **109.27** | 111.95 | 11.20 | **11.32** | 0.41 | **0.41** | 0.54 | **0.56** | 2.75 | **2.98** |
| Glaze | **109.29** | 112.54 | 11.11 | **11.36** | 0.37 | **0.39** | 0.56 | **0.59** | 2.84 | **3.02** |
| SDS(+) | 238.37 | **157.12** | 9.97 | **10.84** | 0.22 | **0.33** | 0.35 | **0.55** | 1.63 | **2.73** |
| SDS(-) | 114.54 | **111.15** | 10.91 | **11.51** | 0.39 | **0.42** | 0.51 | **0.53** | 2.63 | **2.78** |
| SDST | 170.38 | **146.44** | 10.27 | **10.85** | 0.31 | **0.37** | 0.46 | **0.52** | 2.38 | **2.75** |
| *Purification method:IMPRESS* | | | | | | | | | | |
| AntiDB | 312.22 | **167.46** | 9.70 | **11.12** | 0.21 | **0.35** | 0.28 | **0.55** | 1.55 | **2.69** |
| AdvDM(+) | 388.10 | **190.75** | 9.36 | **10.61** | 0.20 | **0.34** | 0.17 | **0.52** | 1.59 | **2.54** |
| AdvDM(-) | 145.07 | **137.96** | 10.87 | **11.58** | 0.40 | **0.43** | 0.31 | **0.39** | 1.76 | **2.19** |
| Glaze | 236.11 | **179.68** | 9.16 | **10.43** | 0.22 | **0.32** | 0.34 | **0.51** | 1.61 | **2.55** |
| SDS(+) | 350.50 | **198.51** | 9.46 | **10.71** | 0.20 | **0.33** | 0.22 | **0.51** | 1.52 | **2.60** |
| SDS(-) | **140.37** | 144.29 | 10.77 | **11.33** | 0.39 | **0.41** | 0.34 | **0.40** | 1.83 | **2.30** |
| SDST | 346.76 | **216.40** | 9.47 | **10.47** | 0.26 | **0.34** | 0.30 | **0.42** | 2.22 | **2.37** |

Under the recommended settings from their respective papers, we measured the additional computation time required by each method compared to the baseline. Using a single NVIDIA RTX A6000 (48GB) GPU, for processing 4 protected images per training task, Noisy-Upscaling required approximately 10 minutes, while IMPRESS needed about 40 minutes. In contrast, IMAP required approximately 30 minutes for masking and patching. However, it is important to note that IMAP's computation time is independent of the size of the training dataset. Therefore, when dealing with large training sets for the same concept, IMAP demonstrates significantly better efficiency than purification-based methods.

Since IMAP and purification-based methods target different stages of customization, they can be used in combination. In Table 3, we present partial experimental results under VGGFace2, which clearly show that applying IMAP after training can substantially enhance the effectiveness of purification-based methods. Furthermore, by comparing the IMAP column in Table 1 with the Purify-Only column in Table 3, we observe that in standalone experiments, IMAP's performance generally falls between that of IMPRESS and Noisy-Upscaling. In Figure 5, we show the visual comparison of different methods under two perturbations for brevity. Given IMAP's high efficiency and flexibility in combining with purification-based methods, we believe IMAP provides a practical and scalable solution for model protection tasks, particularly in scenarios involving large-scale training datasets. Experimental results under CelebA are shown in Appendix D.3.

# 6 CONCLUSION

This paper proposes IMAP, a purification-free framework for restoring diffusion models affected by data perturbations. We investigate how such perturbations alter the relationship between text embeddings and the U-Net backbone in customized diffusion models. IMAP addresses shortcut vulnerabilities through prompt re-mapping, weight masking, and adaptive patching, without requiring clean concept data. Extensive experiments across multiple perturbations and facial datasets demonstrate that IMAP improves distribution consistency and facial clarity of generated images, effectively mitigating the impact of adversarial perturbations. Notably, IMAP achieves higher computational efficiency than purification-based methods—particularly for large-scale datasets—as its runtime is independent of training data size. Moreover, IMAP shows strong compatibility with purification techniques, and their combined use significantly enhances robustness against perturbations.

## AUTHOR STATEMENT ON THE USE OF LLMS

This paper was written by the authors, and a large language model (LLM) was used exclusively for minor edits, grammar checking, and improving the clarity of the text. All ideas, research, and the final content remain the sole intellectual work of the authors.

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

## TECHNICAL APPENDICES

## A  DETAILED MOTIVATION AND PRELIMINARY EXPERIMENTS

Since Liu et al. (2024a) show that adversarial perturbations leads to shortcut learning in customized diffusion models by causing a mismatch between images and prompts in latent space, we build on this idea to study how adversarial perturbations affects the relationship between text embeddings and the U-Net.

**Impact of perturbed fine-tuning on text encoder and U-Net.**

We first generate adversarial data using Anti-DreamBooth (Van Le et al., 2023), then fine-tune for 1000 DreamBooth steps on both clean and adversarial datasets, respectively. As shown in Figure 6, we swap TE and U-Net between the two models and compare the outputs. This lets us isolate the effects of adversarial data on the text encoder (TE) and U-Net.

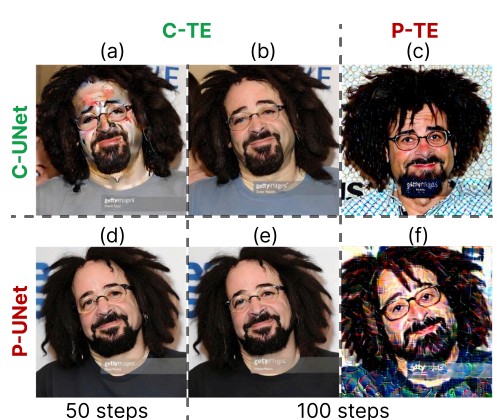

Figure 6: **Sample outputs of mixed text encoder (TE) and U-Net combinations.** C-TE/C-UNet and P-TE/P-UNet refer to models fine-tuned on clean and perturbed data (via Anti-DreamBooth), respectively. Left column shows outputs with 50 denoising steps; middle and right columns show outputs with 100 steps.

Comparing Figure 6 (c) and (f), we find that P-TE with C-UNet still generates clear facial features, but the background shows complex patterns similar to the fully perturbed model (P-TE/P-UNet). This suggests that the perturbed TE embeds noise-like semantics, while the clean U-Net preserves core features and suppresses part of the adversarial effect. In contrast, C-TE with P-UNet produces clean outputs close to the fully clean model (C-TE/C-UNet), showing that P-UNet can still generate well if the text embedding is not adversarially manipulated. This highlights that both TE and U-Net contribute to the final quality. P-TE leverages the generation capacity of C-UNet to inject misleading semantics, leading to background artifacts despite preserved identity. P-UNet further reinforces this shortcut behavior, strengthening the adversarial signal and driving the model away from clean generation.

Interestingly, under the same clean TE, reducing the denoising steps (e.g., from 100 to 50) allows the mixed model (C-TE/P-UNet) to outperform the clean model in both speed and image quality. This suggests that P-UNet learns a rough but stable generation path that converges faster. Similar behavior is observed in adversarial training, where models skip fine details but gain stability. In contrast, C-UNet captures more structure but is sensitive to the initial state, making denoising harder with fewer steps. P-UNet tends to follow shortcut-like recovery paths that are quicker and more robust.

**Localization of adversarially affected U-Net modules.**

To identify the key U-Net modules that contribute most to adversarial effects, we perform another experiment on both clean and perturbed data with frozen TE. We fine-tune only the U-Net and then construct a mixed model (M-UNet) by replacing minimal blocks in the perturbed U-Net (P-UNet) with the corresponding blocks from the clean one (C-UNet). By comparing the outputs, we aim for M-UNet to recover clean-like results. This analysis highlights UpBlock1 and DownBlock2/3, and DownBlock3 as the most impactful blocks. A visual comparison is shown in Figure 7.

C-UNet P-UNet M-UNet  C-UNet P-UNet M-UNet  C-UNet P-UNet M-UNet  C-UNet P-UNet M-UNet

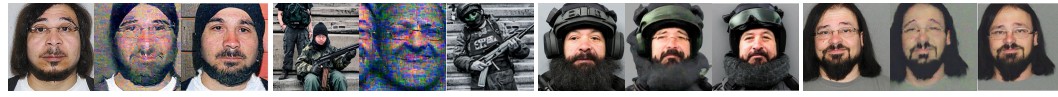

Figure 7: **Comparison of images generated by different U-Net block combinations, with frozen TE during fine-tuning.** C/P/M-UNet refer to clean, perturbed, and mixed models. M-UNet replaces UpBlock1, DownBlock2, and DownBlock3 in P-UNet with the corresponding blocks from C-UNet. The U-Net consists of has four DownBlocks, one MidBlock, and four UpBlocks, indexed from zero.

# B  DETAILS OF EXPERIMENTAL SETTING AND ACCESS TO CODE

## B.1  ACCESS TO CODE AND COMPUTE RESOURCES

An anonymous repository has been provided for code access during the review process:

https://anonymous.4open.science/r/IMAP-anonymous

Although all experiments can be executed on a single NVIDIA RTX A6000 GPU (48 GB), we conduct our experiments using 4 such GPUs to parallelize experiment runs.

For each dataset, the full experimental pipeline takes approximately 28 hours on a single NVIDIA RTX A6000 (48 GB) GPU. This includes training a total of 75 models and sampling 60 images (30 per prompt) for each model: (1 clean fine-tuned model, 7 perturbed baselines, and 7 IMAP-defended models) × 5 identities.

## B.2  DETAILED SETTINGS FOR MODEL CUSTOMIZATION

We use Stable Diffusion V2.1 as pretrained base model. For each experiment based on DreamBooth customization method, we use 4 images as instance dataset, and generate 200 images using $c_{pr}$ as class dataset $D_{pr}$. We set the training step to 1000, the learning rate to 5e-7, and the train batch size to 2 during the fine-tuning stage. All input and generated images have a resolution of 512×512.

## B.3  DETAILED SETTINGS FOR ADVERSARIAL PERTURBATIONS

We focus on the object-driven image synthesis in the context of human face generation, using the VGGFace2 and CelebA-HQ datasets. For each dataset, we randomly choose 5 identities with 12 images per identity. The images are evenly split into three subsets: a clean reference subset, a target subset, and an additional clean reference. Base on these subsets, we conduct the perturbed dataset with 4 images per identity, with the original target subset as clean dataset.

All adversarial perturbation methods are evaluated under a fixed perturbation budget of $\delta = 16/255$. As Glaze does not provide access to its source code and does not allow control over the perturbation budget, we use its maximum perturbation setting.

## B.4  DETAILED SETTINGS FOR IMAP

In the weight masking stage, we generate 20 images per prompt for DDIM inversion, using 50 inversion steps. We compute the mean of the feature maps across timesteps 10 to 20, as the later steps are essentially close to random noise.

In the adaptive patching stage, we use the dataset $D_{pt}$, consisting of 200 images generated by the pretrained model with the prompt $c^B$-"a photo of noised person"- as the instance dataset. The class dataset $D_{pr}$ is the same as that used in the fine-tuning stage. The patching step is set to 1000, with a learning rate of 5e-8.

## C  MORE VISUAL COMPARISONS OF GENERATED IMAGES

We provide additional visual comparisons of the generated images under different sampling prompts between the baseline model and our IMAP-defended model. Samples under the $c_{train}$ prompt are shown in Figure 8, while those under the $c_{diff}$ prompt are shown in Figure 9.

These comparisons further illustrate the robustness and visual fidelity achieved by our proposed defense.

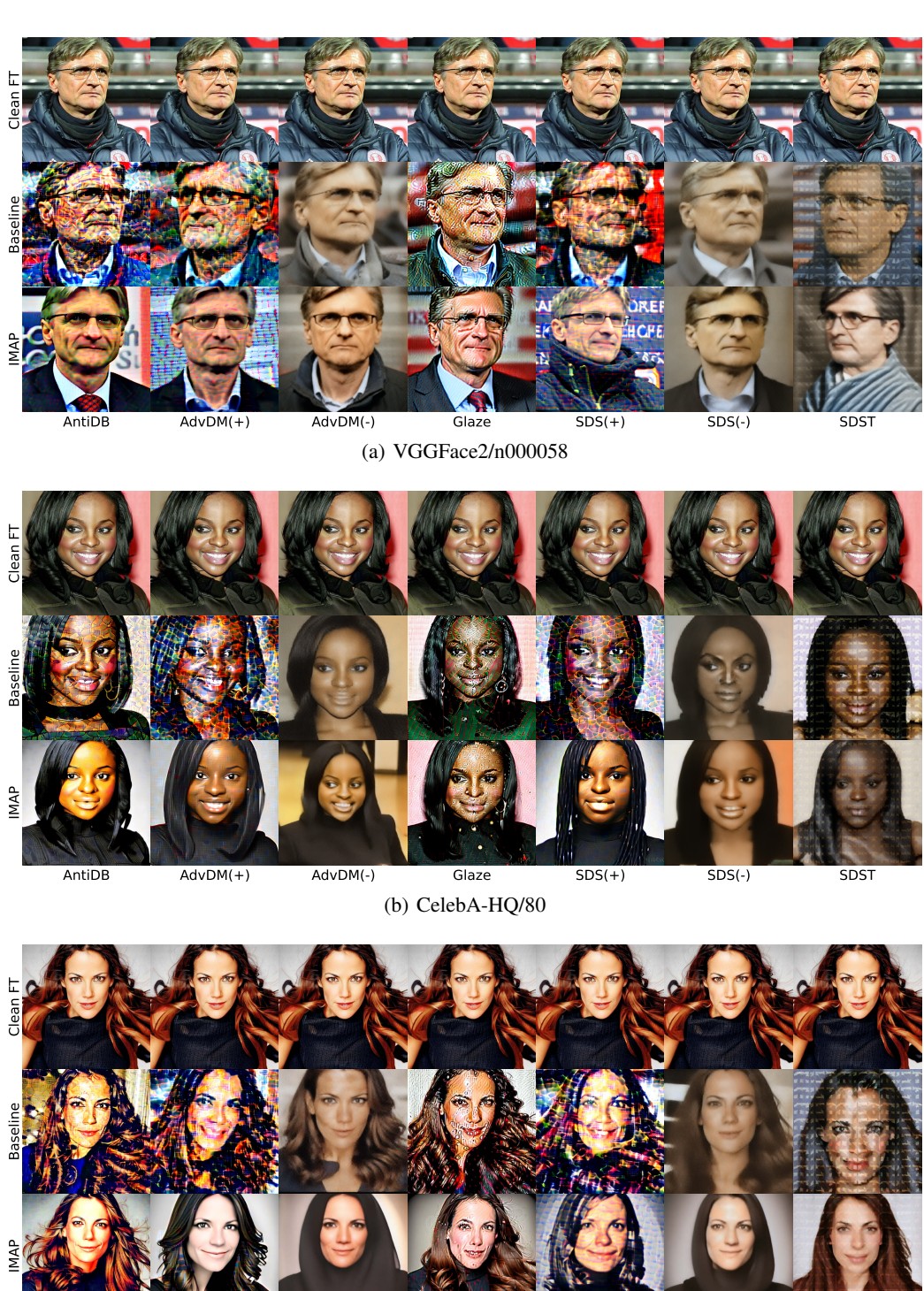

(a) VGGFace2/n000058

(b) CelebA-HQ/80

(c) CelebA-HQ/108

Figure 8: **Visual comparison of images generated by customized diffusion models under different adversarial perturbations and data identities using prompt** $c_{train}$**.** Each subfigure is titled in the format *dataset/data-identity*. In each subfigure, the top row shows results from the clean fine-tuned model, the middle row shows results from the baseline model, and the bottom row shows results from the IMAP-defended model. Each column corresponds to a different adversarial perturbation method.

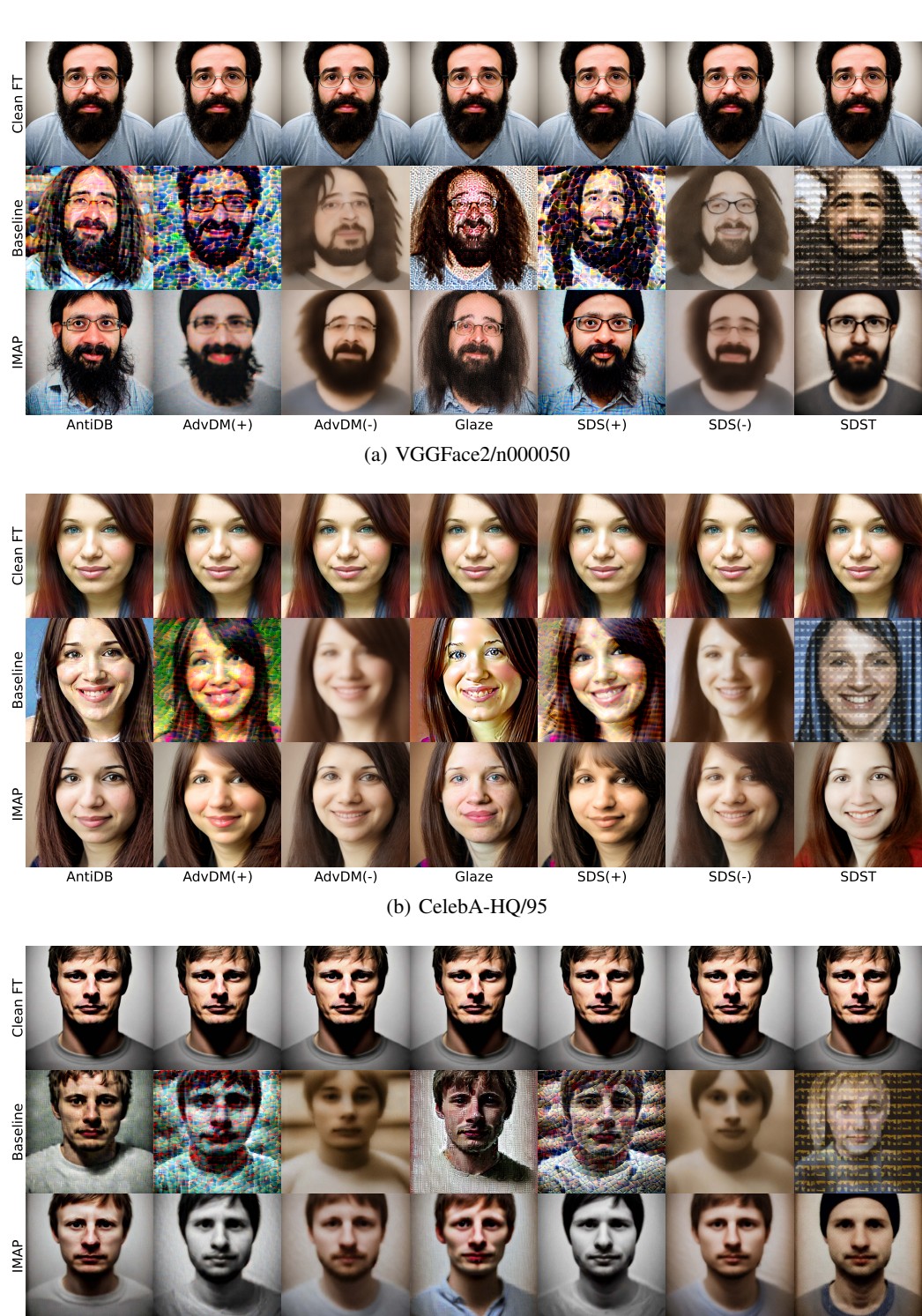

(a) VGGFace2/n000050

(b) CelebA-HQ/95

(c) CelebA-HQ/103

Figure 9: **Visual comparison of images generated by customized diffusion models under different adversarial perturbations and data identities using prompt $c_{diff}$.**

# D  SUPPLEMENTARY EXPERIMENTAL RESULTS

## D.1  ERROR BARS

Due to customizing DMs on different identities for each dataset, the results may exhibit variations. Thus, we conduct independent experiments under same settings for each identity, and then calculate the mean and the standard deviation for each evaluated metric per dataset.

For each metric to be evaluated, we perform independent experiment under same settings and random seed on N = 5 data identities for each dataset, obtaining N result values $v_1, v_2, ..., v_N$. The mean $\mu$ and the standard deviation $\sigma$ of these values are calculated as follows:

$$\mu = \frac{1}{N} \sum_{i=1}^{N} v_i \tag{8}$$

$$\sigma = \sqrt{\frac{1}{N-1} \sum_{i=1}^{N} (v_i - \mu)^2} \tag{9}$$

## D.2  SUPPLEMENTARY RESULTS FOR SECTION 5.2

We provide a more detailed quantitative evaluation with error bars, as shown in Table 4 to Table 13.

Specifically, the experiments from Table 4 to Table 8 correspond to the VGGFace2 dataset, and the experiments from Table 9 to Table 13 correspond to the CelebA-HQ dataset. In the tables, we provide a quantitative evaluation of the images generated by the baseline model and the IMAP-defended model, compared to the clean fine-tuning model, under different metrics for each data identity and perturbation.

Table 4: **Quantitative evaluation on VGGFace2 under different sampling prompt settings (FID↓).**

| Adersarial Perturbations | Dataset: VGGFace2, Metric: FID↓ | | | | | | | |
|---|---|---|---|---|---|---|---|---|
| | Identity | n000050 | n000057 | n000058 | n000063 | n000138 | mean | std |
| *Both prompts: $c_{train}, c_{diff}$* | | | | | | | | |
| AntiDB | baseline | 346.13 | 436.45 | 345.46 | 373.57 | 387.60 | 377.84 | 37.42 |
| | IMAP | 159.83 | 124.27 | 128.33 | 212.96 | 171.67 | 159.41 | 36.12 |
| AdvDM(+) | baseline | 411.30 | 437.97 | 361.70 | 413.97 | 361.72 | 397.33 | 34.14 |
| | IMAP | 170.72 | 154.72 | 166.23 | 208.13 | 224.34 | 184.83 | 29.81 |
| AdvDM(-) | baseline | 158.14 | 122.13 | 155.84 | 173.38 | 130.44 | 147.99 | 21.13 |
| | IMAP | 145.29 | 122.02 | 111.25 | 161.41 | 142.18 | 136.43 | 19.86 |
| Glaze | baseline | 271.04 | 176.92 | 216.95 | 246.48 | 211.18 | 224.52 | 35.87 |
| | IMAP | 202.16 | 126.76 | 165.36 | 199.45 | 160.85 | 170.92 | 31.11 |
| SDS(+) | baseline | 363.34 | 411.69 | 364.44 | 376.15 | 378.68 | 378.86 | 19.58 |
| | IMAP | 155.11 | 161.73 | 176.57 | 209.58 | 213.09 | 183.22 | 26.85 |
| SDS(-) | baseline | 169.26 | 122.73 | 145.04 | 172.03 | 129.31 | 147.67 | 22.50 |
| | IMAP | 145.81 | 118.38 | 132.38 | 174.15 | 143.79 | 142.90 | 20.60 |
| SDST | baseline | 329.68 | 299.14 | 363.41 | 320.32 | 324.47 | 327.41 | 23.23 |
| | IMAP | 194.19 | 158.66 | 163.89 | 248.75 | 232.56 | 199.61 | 40.26 |
| *Train prompt: $c_{train}$* | | | | | | | | |
| AntiDB | baseline | 428.43 | 469.25 | 440.37 | 445.30 | 411.56 | 438.98 | 21.34 |
| | IMAP | 180.41 | 181.47 | 174.64 | 283.27 | 212.09 | 206.38 | 45.41 |
| AdvDM(+) | baseline | 444.74 | 472.07 | 436.34 | 475.97 | 436.07 | 453.04 | 19.52 |
| | IMAP | 200.39 | 227.58 | 268.08 | 270.30 | 298.69 | 253.01 | 38.82 |
| AdvDM(-) | baseline | 174.67 | 137.70 | 166.71 | 195.02 | 163.90 | 167.60 | 20.68 |
| | IMAP | 182.59 | 148.32 | 159.60 | 234.92 | 207.25 | 186.53 | 35.26 |
| Glaze | baseline | 236.26 | 190.02 | 221.05 | 267.64 | 258.59 | 234.71 | 30.99 |
| | IMAP | 239.78 | 175.06 | 235.17 | 251.36 | 223.02 | 224.88 | 29.64 |
| SDS(+) | baseline | 392.33 | 468.60 | 418.05 | 439.47 | 424.68 | 428.63 | 28.11 |
| | IMAP | 186.12 | 253.12 | 271.76 | 290.18 | 291.30 | 258.50 | 43.37 |
| SDS(-) | baseline | 190.56 | 134.91 | 171.12 | 189.96 | 177.87 | 172.89 | 22.77 |
| | IMAP | 182.12 | 150.11 | 190.90 | 226.22 | 189.87 | 187.84 | 27.12 |
| SDST | baseline | 292.33 | 298.78 | 386.70 | 358.38 | 327.36 | 332.71 | 39.95 |
| | IMAP | 232.92 | 226.17 | 214.05 | 335.21 | 338.67 | 269.40 | 62.03 |
| *Diff prompt: $c_{diff}$* | | | | | | | | |
| AntiDB | baseline | 320.13 | 440.80 | 287.71 | 358.75 | 399.24 | 361.33 | 60.98 |
| | IMAP | 195.78 | 108.91 | 124.64 | 218.08 | 177.81 | 165.05 | 46.65 |
| AdvDM(+) | baseline | 415.90 | 435.79 | 332.90 | 390.97 | 324.67 | 380.05 | 49.50 |
| | IMAP | 192.39 | 131.34 | 108.42 | 208.30 | 194.36 | 166.96 | 44.16 |
| AdvDM(-) | baseline | 186.30 | 144.32 | 191.00 | 200.54 | 148.65 | 174.16 | 25.83 |
| | IMAP | 144.33 | 128.98 | 103.47 | 145.73 | 134.78 | 131.46 | 17.10 |
| Glaze | baseline | 345.80 | 200.73 | 250.96 | 285.48 | 203.26 | 257.25 | 60.80 |
| | IMAP | 204.98 | 138.47 | 141.95 | 198.17 | 141.01 | 164.92 | 33.58 |
| SDS(+) | baseline | 371.00 | 389.44 | 353.05 | 348.29 | 377.07 | 367.77 | 17.05 |
| | IMAP | 169.44 | 127.03 | 126.53 | 189.72 | 189.48 | 160.44 | 31.81 |
| SDS(-) | baseline | 184.08 | 139.49 | 157.70 | 195.37 | 134.64 | 162.26 | 26.81 |
| | IMAP | 146.89 | 123.70 | 115.57 | 170.38 | 143.73 | 140.05 | 21.49 |
| SDST | baseline | 415.33 | 343.93 | 391.02 | 326.22 | 355.56 | 366.41 | 36.18 |
| | IMAP | 198.11 | 145.70 | 166.79 | 215.40 | 167.10 | 178.62 | 27.79 |

Table 5: **Quantitative evaluation on VGGFace2 under different sampling prompt settings (PSNR↑).**

| Adersarial Perturbations | | Dataset: VGGFace2, Metric: PSNR↑ | | | | | | |
|---|---|---|---|---|---|---|---|---|
| | Identity | n000050 | n000057 | n000058 | n000063 | n000138 | mean | std |
| *Both prompts: $c_{train}, c_{diff}$* | | | | | | | | |
| AntiDB | baseline | 9.95 | 9.43 | 9.46 | 9.09 | 8.97 | 9.38 | 0.38 |
| | IMAP | 10.84 | 12.12 | 10.66 | 10.11 | 11.07 | 10.96 | 0.74 |
| AdvDM(+) | baseline | 8.89 | 9.11 | 9.40 | 8.57 | 9.25 | 9.05 | 0.33 |
| | IMAP | 10.27 | 11.39 | 10.95 | 10.12 | 11.06 | 10.76 | 0.54 |
| AdvDM(-) | baseline | 10.96 | 10.76 | 11.56 | 10.69 | 11.64 | 11.12 | 0.45 |
| | IMAP | 12.02 | 11.61 | 12.27 | 11.02 | 11.68 | 11.72 | 0.47 |
| Glaze | baseline | 8.84 | 9.65 | 8.93 | 8.38 | 9.75 | 9.11 | 0.58 |
| | IMAP | 10.40 | 10.64 | 10.29 | 9.85 | 10.76 | 10.39 | 0.35 |
| SDS(+) | baseline | 9.36 | 9.67 | 9.79 | 8.82 | 9.27 | 9.38 | 0.38 |
| | IMAP | 10.98 | 11.51 | 10.70 | 10.44 | 10.68 | 10.86 | 0.41 |
| SDS(-) | baseline | 10.76 | 10.79 | 11.71 | 10.66 | 11.60 | 11.10 | 0.51 |
| | IMAP | 11.98 | 12.27 | 12.35 | 10.95 | 11.37 | 11.78 | 0.61 |
| SDST | baseline | 9.47 | 10.59 | 11.11 | 9.38 | 9.54 | 10.02 | 0.78 |
| | IMAP | 10.61 | 11.73 | 11.23 | 10.03 | 10.98 | 10.92 | 0.64 |
| *Train prompt: $c_{train}$* | | | | | | | | |
| AntiDB | baseline | 8.96 | 9.35 | 8.76 | 8.54 | 8.59 | 8.84 | 0.33 |
| | IMAP | 9.37 | 10.74 | 9.39 | 8.89 | 9.97 | 9.67 | 0.71 |
| AdvDM(+) | baseline | 8.42 | 8.77 | 8.94 | 8.34 | 8.49 | 8.59 | 0.25 |
| | IMAP | 8.73 | 9.93 | 9.48 | 9.36 | 9.72 | 9.45 | 0.45 |
| AdvDM(-) | baseline | 11.05 | 10.66 | 11.35 | 10.32 | 11.23 | 10.92 | 0.43 |
| | IMAP | 10.73 | 10.79 | 10.99 | 9.60 | 10.26 | 10.47 | 0.56 |
| Glaze | baseline | 8.64 | 9.13 | 8.85 | 7.59 | 8.73 | 8.58 | 0.59 |
| | IMAP | 8.49 | 9.26 | 8.63 | 8.53 | 9.22 | 8.83 | 0.38 |
| SDS(+) | baseline | 8.79 | 9.62 | 9.12 | 8.43 | 8.30 | 8.85 | 0.54 |
| | IMAP | 9.60 | 9.86 | 9.55 | 9.48 | 9.41 | 9.58 | 0.17 |
| SDS(-) | baseline | 10.32 | 10.56 | 11.41 | 10.02 | 11.08 | 10.68 | 0.56 |
| | IMAP | 10.44 | 11.61 | 11.16 | 9.89 | 10.25 | 10.67 | 0.70 |
| SDST | baseline | 9.06 | 10.86 | 11.54 | 9.21 | 9.33 | 10.00 | 1.12 |
| | IMAP | 9.59 | 11.12 | 11.04 | 8.70 | 9.79 | 10.05 | 1.03 |
| *Diff prompt: $c_{diff}$* | | | | | | | | |
| AntiDB | baseline | 10.94 | 9.52 | 10.15 | 9.65 | 9.36 | 9.92 | 0.64 |
| | IMAP | 12.31 | 13.49 | 11.93 | 11.33 | 12.16 | 12.24 | 0.79 |
| AdvDM(+) | baseline | 9.36 | 9.46 | 9.87 | 8.80 | 10.01 | 9.50 | 0.48 |
| | IMAP | 11.81 | 12.84 | 12.42 | 10.87 | 12.40 | 12.07 | 0.76 |
| AdvDM(-) | baseline | 10.87 | 10.86 | 11.76 | 11.05 | 12.05 | 11.32 | 0.55 |
| | IMAP | 13.31 | 12.42 | 13.55 | 12.44 | 13.11 | 12.97 | 0.51 |
| Glaze | baseline | 9.05 | 10.17 | 9.02 | 9.17 | 10.78 | 9.64 | 0.80 |
| | IMAP | 12.31 | 12.01 | 11.95 | 11.17 | 12.29 | 11.95 | 0.47 |
| SDS(+) | baseline | 9.94 | 9.72 | 10.47 | 9.21 | 10.24 | 9.92 | 0.49 |
| | IMAP | 12.37 | 13.16 | 11.85 | 11.40 | 11.94 | 12.15 | 0.66 |
| SDS(-) | baseline | 11.21 | 11.01 | 12.02 | 11.29 | 12.12 | 11.53 | 0.50 |
| | IMAP | 13.51 | 12.94 | 13.55 | 12.01 | 12.48 | 12.90 | 0.67 |
| SDST | baseline | 9.88 | 10.32 | 10.68 | 9.54 | 9.74 | 10.04 | 0.46 |
| | IMAP | 11.63 | 12.33 | 11.42 | 11.36 | 12.17 | 11.78 | 0.44 |

Table 6: **Quantitative evaluation on VGGFace2 under different sampling prompt settings (SSIM↑).**

| Adersarial Perturbations | | Dataset: VGGFace2, Metric: SSIM↑ | | | | | | |
|---|---|---|---|---|---|---|---|---|
| | Identity | n000050 | n000057 | n000058 | n000063 | n000138 | mean | std |
| *Both prompts:* $c_{train}, c_{diff}$ | | | | | | | | |
| AntiDB | baseline | 0.23 | 0.19 | 0.21 | 0.21 | 0.20 | 0.21 | 0.02 |
| | IMAP | 0.33 | 0.40 | 0.32 | 0.34 | 0.38 | 0.35 | 0.04 |
| AdvDM(+) | baseline | 0.18 | 0.19 | 0.23 | 0.20 | 0.22 | 0.20 | 0.02 |
| | IMAP | 0.31 | 0.36 | 0.32 | 0.35 | 0.36 | 0.34 | 0.02 |
| AdvDM(-) | baseline | 0.34 | 0.38 | 0.38 | 0.37 | 0.46 | 0.38 | 0.04 |
| | IMAP | 0.40 | 0.41 | 0.41 | 0.42 | 0.49 | 0.42 | 0.04 |
| Glaze | baseline | 0.16 | 0.23 | 0.17 | 0.17 | 0.30 | 0.21 | 0.06 |
| | IMAP | 0.28 | 0.31 | 0.28 | 0.32 | 0.35 | 0.31 | 0.03 |
| SDS(+) | baseline | 0.20 | 0.21 | 0.22 | 0.20 | 0.23 | 0.21 | 0.01 |
| | IMAP | 0.35 | 0.34 | 0.30 | 0.39 | 0.37 | 0.35 | 0.03 |
| SDS(-) | baseline | 0.33 | 0.37 | 0.38 | 0.37 | 0.46 | 0.38 | 0.05 |
| | IMAP | 0.39 | 0.42 | 0.41 | 0.41 | 0.48 | 0.42 | 0.03 |
| SDST | baseline | 0.23 | 0.27 | 0.29 | 0.27 | 0.34 | 0.28 | 0.04 |
| | IMAP | 0.33 | 0.37 | 0.35 | 0.33 | 0.43 | 0.36 | 0.04 |
| *Train prompt:* $c_{train}$ | | | | | | | | |
| AntiDB | baseline | 0.17 | 0.18 | 0.19 | 0.19 | 0.19 | 0.18 | 0.01 |
| | IMAP | 0.22 | 0.30 | 0.25 | 0.25 | 0.30 | 0.27 | 0.04 |
| AdvDM(+) | baseline | 0.15 | 0.17 | 0.20 | 0.18 | 0.20 | 0.18 | 0.02 |
| | IMAP | 0.21 | 0.26 | 0.21 | 0.30 | 0.27 | 0.25 | 0.04 |
| AdvDM(-) | baseline | 0.29 | 0.35 | 0.37 | 0.32 | 0.42 | 0.35 | 0.05 |
| | IMAP | 0.30 | 0.33 | 0.35 | 0.35 | 0.43 | 0.35 | 0.05 |
| Glaze | baseline | 0.13 | 0.16 | 0.17 | 0.12 | 0.22 | 0.16 | 0.04 |
| | IMAP | 0.15 | 0.19 | 0.17 | 0.21 | 0.25 | 0.19 | 0.04 |
| SDS(+) | baseline | 0.17 | 0.20 | 0.19 | 0.19 | 0.21 | 0.19 | 0.01 |
| | IMAP | 0.26 | 0.20 | 0.21 | 0.34 | 0.27 | 0.25 | 0.06 |
| SDS(-) | baseline | 0.27 | 0.33 | 0.36 | 0.33 | 0.43 | 0.34 | 0.06 |
| | IMAP | 0.29 | 0.35 | 0.36 | 0.35 | 0.43 | 0.36 | 0.05 |
| SDST | baseline | 0.21 | 0.25 | 0.31 | 0.25 | 0.31 | 0.26 | 0.04 |
| | IMAP | 0.24 | 0.28 | 0.32 | 0.24 | 0.35 | 0.29 | 0.05 |
| *Diff prompt:* $c_{diff}$ | | | | | | | | |
| AntiDB | baseline | 0.30 | 0.20 | 0.23 | 0.24 | 0.20 | 0.24 | 0.04 |
| | IMAP | 0.44 | 0.50 | 0.39 | 0.43 | 0.46 | 0.44 | 0.04 |
| AdvDM(+) | baseline | 0.20 | 0.21 | 0.26 | 0.21 | 0.24 | 0.23 | 0.02 |
| | IMAP | 0.42 | 0.46 | 0.42 | 0.40 | 0.46 | 0.43 | 0.03 |
| AdvDM(-) | baseline | 0.38 | 0.41 | 0.38 | 0.42 | 0.50 | 0.42 | 0.05 |
| | IMAP | 0.49 | 0.48 | 0.47 | 0.49 | 0.55 | 0.50 | 0.03 |
| Glaze | baseline | 0.18 | 0.30 | 0.18 | 0.22 | 0.39 | 0.25 | 0.09 |
| | IMAP | 0.40 | 0.44 | 0.38 | 0.42 | 0.46 | 0.42 | 0.03 |
| SDS(+) | baseline | 0.23 | 0.22 | 0.24 | 0.22 | 0.25 | 0.23 | 0.01 |
| | IMAP | 0.43 | 0.48 | 0.40 | 0.43 | 0.48 | 0.44 | 0.04 |
| SDS(-) | baseline | 0.38 | 0.41 | 0.39 | 0.42 | 0.50 | 0.42 | 0.05 |
| | IMAP | 0.49 | 0.49 | 0.46 | 0.47 | 0.53 | 0.49 | 0.03 |
| SDST | baseline | 0.26 | 0.29 | 0.28 | 0.30 | 0.36 | 0.30 | 0.04 |
| | IMAP | 0.42 | 0.46 | 0.39 | 0.43 | 0.50 | 0.44 | 0.04 |

Table 7: **Quantitative evaluation on VGGFace2 under different sampling prompt settings (TOPIQ↑).**

| Adersarial Perturbations | | Dataset: CelebA-HQ, Metric: TOPIQ↑ | | | | | | |
|---|---|---|---|---|---|---|---|---|
| | Identity | n000050 | n000057 | n000058 | n000063 | n000138 | mean | std |
| Both prompts: $c_{train}, c_{diff}$ | | | | | | | | |
| AntiDB | baseline | 0.23 | 0.03 | 0.24 | 0.28 | 0.20 | 0.20 | 0.10 |
| | IMAP | 0.71 | 0.56 | 0.56 | 0.49 | 0.50 | 0.56 | 0.09 |
| AdvDM(+) | baseline | 0.13 | 0.02 | 0.15 | 0.09 | 0.24 | 0.13 | 0.08 |
| | IMAP | 0.58 | 0.54 | 0.58 | 0.52 | 0.48 | 0.54 | 0.04 |
| AdvDM(-) | baseline | 0.42 | 0.43 | 0.28 | 0.30 | 0.29 | 0.34 | 0.07 |
| | IMAP | 0.43 | 0.49 | 0.49 | 0.47 | 0.40 | 0.46 | 0.04 |
| Glaze | baseline | 0.23 | 0.46 | 0.38 | 0.30 | 0.35 | 0.34 | 0.09 |
| | IMAP | 0.55 | 0.53 | 0.49 | 0.52 | 0.48 | 0.51 | 0.03 |
| SDS(+) | baseline | 0.15 | 0.03 | 0.14 | 0.22 | 0.26 | 0.16 | 0.09 |
| | IMAP | 0.69 | 0.51 | 0.53 | 0.54 | 0.47 | 0.55 | 0.08 |
| SDS(-) | baseline | 0.36 | 0.36 | 0.31 | 0.27 | 0.31 | 0.32 | 0.04 |
| | IMAP | 0.46 | 0.38 | 0.40 | 0.35 | 0.38 | 0.39 | 0.04 |
| SDST | baseline | 0.35 | 0.41 | 0.31 | 0.35 | 0.35 | 0.35 | 0.04 |
| | IMAP | 0.55 | 0.49 | 0.44 | 0.49 | 0.37 | 0.47 | 0.07 |
| Train prompt: $c_{train}$ | | | | | | | | |
| AntiDB | baseline | 0.22 | 0.00 | 0.14 | 0.17 | 0.06 | 0.12 | 0.09 |
| | IMAP | 0.65 | 0.54 | 0.44 | 0.34 | 0.37 | 0.47 | 0.13 |
| AdvDM(+) | baseline | 0.22 | 0.00 | 0.04 | 0.04 | 0.07 | 0.07 | 0.08 |
| | IMAP | 0.47 | 0.44 | 0.43 | 0.38 | 0.37 | 0.42 | 0.04 |
| AdvDM(-) | baseline | 0.35 | 0.43 | 0.30 | 0.34 | 0.30 | 0.35 | 0.06 |
| | IMAP | 0.27 | 0.47 | 0.35 | 0.42 | 0.36 | 0.38 | 0.07 |
| Glaze | baseline | 0.28 | 0.36 | 0.35 | 0.23 | 0.25 | 0.30 | 0.06 |
| | IMAP | 0.35 | 0.38 | 0.25 | 0.34 | 0.33 | 0.33 | 0.05 |
| SDS(+) | baseline | 0.12 | 0.01 | 0.01 | 0.20 | 0.25 | 0.12 | 0.11 |
| | IMAP | 0.60 | 0.31 | 0.36 | 0.42 | 0.32 | 0.40 | 0.12 |
| SDS(-) | baseline | 0.33 | 0.39 | 0.33 | 0.29 | 0.31 | 0.33 | 0.04 |
| | IMAP | 0.33 | 0.37 | 0.33 | 0.33 | 0.34 | 0.34 | 0.02 |
| SDST | baseline | 0.40 | 0.45 | 0.33 | 0.39 | 0.38 | 0.39 | 0.04 |
| | IMAP | 0.42 | 0.40 | 0.37 | 0.46 | 0.24 | 0.38 | 0.08 |
| Diff prompt: $c_{diff}$ | | | | | | | | |
| AntiDB | baseline | 0.24 | 0.06 | 0.33 | 0.39 | 0.35 | 0.27 | 0.13 |
| | IMAP | 0.78 | 0.58 | 0.67 | 0.64 | 0.63 | 0.66 | 0.07 |
| AdvDM(+) | baseline | 0.04 | 0.04 | 0.25 | 0.14 | 0.41 | 0.18 | 0.16 |
| | IMAP | 0.69 | 0.65 | 0.74 | 0.65 | 0.59 | 0.66 | 0.05 |
| AdvDM(-) | baseline | 0.49 | 0.42 | 0.26 | 0.27 | 0.28 | 0.34 | 0.10 |
| | IMAP | 0.59 | 0.52 | 0.63 | 0.51 | 0.43 | 0.53 | 0.08 |
| Glaze | baseline | 0.18 | 0.57 | 0.40 | 0.37 | 0.45 | 0.39 | 0.14 |
| | IMAP | 0.75 | 0.69 | 0.73 | 0.70 | 0.63 | 0.70 | 0.04 |
| SDS(+) | baseline | 0.19 | 0.04 | 0.26 | 0.24 | 0.27 | 0.20 | 0.09 |
| | IMAP | 0.77 | 0.70 | 0.70 | 0.66 | 0.63 | 0.69 | 0.06 |
| SDS(-) | baseline | 0.40 | 0.33 | 0.30 | 0.25 | 0.32 | 0.32 | 0.05 |
| | IMAP | 0.58 | 0.39 | 0.48 | 0.37 | 0.42 | 0.45 | 0.08 |
| SDST | baseline | 0.31 | 0.37 | 0.29 | 0.30 | 0.32 | 0.32 | 0.03 |
| | IMAP | 0.69 | 0.59 | 0.52 | 0.51 | 0.50 | 0.56 | 0.08 |

Table 8: **Quantitative evaluation on VGGFace2 under different sampling prompt settings (QAlign↑).**

| Adersarial Perturbations | | Dataset: CelebA-HQ, Metric: QAlign↑ | | | | | | |
|---|---|---|---|---|---|---|---|---|
| | Identity | n000050 | n000057 | n000058 | n000063 | n000138 | mean | std |
| *Both prompts: $c_{train}, c_{diff}$* | | | | | | | | |
| AntiDB | baseline | 1.28 | 1.58 | 1.33 | 1.80 | 1.77 | 1.55 | 0.24 |
| | IMAP | 3.38 | 2.65 | 2.54 | 2.61 | 2.56 | 2.75 | 0.36 |
| AdvDM(+) | baseline | 1.43 | 1.35 | 1.43 | 1.32 | 1.64 | 1.43 | 0.13 |
| | IMAP | 2.63 | 2.43 | 2.49 | 2.63 | 2.21 | 2.48 | 0.17 |
| AdvDM(-) | baseline | 2.04 | 2.12 | 1.67 | 1.99 | 2.03 | 1.97 | 0.17 |
| | IMAP | 2.45 | 2.49 | 2.83 | 2.61 | 2.56 | 2.59 | 0.15 |
| Glaze | baseline | 1.41 | 1.92 | 1.47 | 1.56 | 1.63 | 1.60 | 0.20 |
| | IMAP | 2.74 | 2.51 | 2.46 | 2.79 | 2.43 | 2.59 | 0.16 |
| SDS(+) | baseline | 1.44 | 1.44 | 1.45 | 1.49 | 1.63 | 1.49 | 0.08 |
| | IMAP | 3.30 | 2.63 | 2.51 | 3.08 | 2.52 | 2.81 | 0.36 |
| SDS(-) | baseline | 1.87 | 1.94 | 1.83 | 1.85 | 2.04 | 1.91 | 0.09 |
| | IMAP | 2.53 | 2.12 | 2.32 | 2.14 | 2.42 | 2.31 | 0.18 |
| SDST | baseline | 1.91 | 2.50 | 1.95 | 2.36 | 1.88 | 2.12 | 0.29 |
| | IMAP | 2.92 | 2.81 | 2.20 | 2.52 | 2.28 | 2.54 | 0.32 |
| *Train prompt: $c_{train}$* | | | | | | | | |
| AntiDB | baseline | 1.26 | 1.60 | 1.29 | 1.75 | 1.63 | 1.51 | 0.22 |
| | IMAP | 2.87 | 2.24 | 1.90 | 1.79 | 1.94 | 2.15 | 0.44 |
| AdvDM(+) | baseline | 1.45 | 1.33 | 1.43 | 1.33 | 1.55 | 1.42 | 0.09 |
| | IMAP | 1.92 | 1.72 | 1.52 | 1.87 | 1.54 | 1.72 | 0.19 |
| AdvDM(-) | baseline | 1.86 | 2.14 | 1.79 | 2.19 | 2.22 | 2.04 | 0.20 |
| | IMAP | 1.74 | 2.31 | 2.19 | 2.34 | 2.55 | 2.23 | 0.30 |
| Glaze | baseline | 1.55 | 1.49 | 1.31 | 1.19 | 1.41 | 1.39 | 0.15 |
| | IMAP | 1.86 | 1.57 | 1.28 | 1.79 | 1.75 | 1.65 | 0.23 |
| SDS(+) | baseline | 1.43 | 1.36 | 1.42 | 1.44 | 1.50 | 1.43 | 0.05 |
| | IMAP | 2.68 | 1.73 | 1.57 | 2.44 | 1.65 | 2.02 | 0.51 |
| SDS(-) | baseline | 1.72 | 2.01 | 1.88 | 1.93 | 2.17 | 1.94 | 0.17 |
| | IMAP | 1.97 | 2.05 | 2.01 | 2.05 | 2.41 | 2.10 | 0.18 |
| SDST | baseline | 2.22 | 2.71 | 1.74 | 2.58 | 2.02 | 2.25 | 0.40 |
| | IMAP | 2.15 | 2.52 | 1.73 | 2.28 | 1.72 | 2.08 | 0.35 |
| *Diff prompt: $c_{diff}$* | | | | | | | | |
| AntiDB | baseline | 1.30 | 1.56 | 1.38 | 1.85 | 1.91 | 1.60 | 0.27 |
| | IMAP | 3.90 | 3.05 | 3.18 | 3.44 | 3.18 | 3.35 | 0.34 |
| AdvDM(+) | baseline | 1.42 | 1.36 | 1.44 | 1.31 | 1.73 | 1.45 | 0.16 |
| | IMAP | 3.33 | 3.14 | 3.46 | 3.39 | 2.88 | 3.24 | 0.23 |
| AdvDM(-) | baseline | 2.23 | 2.10 | 1.55 | 1.79 | 1.84 | 1.90 | 0.27 |
| | IMAP | 3.15 | 2.67 | 3.46 | 2.87 | 2.59 | 2.95 | 0.36 |
| Glaze | baseline | 1.28 | 2.35 | 1.64 | 1.93 | 1.86 | 1.81 | 0.39 |
| | IMAP | 3.62 | 3.45 | 3.65 | 3.77 | 3.12 | 3.52 | 0.25 |
| SDS(+) | baseline | 1.46 | 1.51 | 1.47 | 1.54 | 1.76 | 1.55 | 0.12 |
| | IMAP | 3.91 | 3.53 | 3.46 | 3.72 | 3.38 | 3.60 | 0.21 |
| SDS(-) | baseline | 2.03 | 1.88 | 1.78 | 1.77 | 1.92 | 1.87 | 0.11 |
| | IMAP | 3.09 | 2.18 | 2.62 | 2.22 | 2.44 | 2.51 | 0.37 |
| SDST | baseline | 1.59 | 2.30 | 2.15 | 2.14 | 1.73 | 1.98 | 0.30 |
| | IMAP | 3.69 | 3.10 | 2.66 | 2.76 | 2.84 | 3.01 | 0.41 |

Table 9: **Quantitative evaluation on CelebA-HQ under different sampling prompt settings (FID↓).**

| Adersarial Perturbations | | Dataset: CelebA-HQ, Metric: FID↓ | | | | | | | |
|---|---|---|---|---|---|---|---|---|---|
| | Identity | 80 | 95 | 103 | 104 | 108 | mean | std |
| *Both prompts: $c_{train}, c_{diff}$* | | | | | | | | |
| AntiDB | baseline | 201.92 | 230.60 | 258.97 | 184.73 | 255.02 | 226.25 | 32.53 |
| | IMAP | 142.38 | 140.74 | 93.48 | 154.24 | 157.07 | 137.58 | 25.67 |
| AdvDM(+) | baseline | 369.41 | 434.86 | 449.22 | 318.32 | 364.74 | 387.31 | 54.05 |
| | IMAP | 157.74 | 202.75 | 189.04 | 166.21 | 197.86 | 182.72 | 19.79 |
| AdvDM(-) | baseline | 167.90 | 128.37 | 127.52 | 119.97 | 145.10 | 137.77 | 19.18 |
| | IMAP | 153.49 | 157.17 | 98.51 | 132.71 | 131.57 | 134.69 | 23.35 |
| Glaze | baseline | 187.28 | 237.16 | 249.07 | 175.00 | 226.62 | 215.03 | 32.23 |
| | IMAP | 170.77 | 183.27 | 160.14 | 167.13 | 178.00 | 171.86 | 9.07 |
| SDS(+) | baseline | 315.58 | 396.82 | 397.79 | 342.26 | 384.15 | 367.32 | 36.70 |
| | IMAP | 164.14 | 208.39 | 96.32 | 131.01 | 211.84 | 162.34 | 49.79 |
| SDS(-) | baseline | 168.88 | 119.63 | 107.04 | 119.76 | 147.45 | 132.55 | 25.12 |
| | IMAP | 159.53 | 137.07 | 99.79 | 135.70 | 137.65 | 133.95 | 21.50 |
| SDST | baseline | 283.22 | 275.57 | 303.98 | 206.29 | 271.42 | 268.10 | 36.75 |
| | IMAP | 181.17 | 172.38 | 150.92 | 164.40 | 153.56 | 164.48 | 12.69 |
| *Train prompt: $c_{train}$* | | | | | | | | |
| AntiDB | baseline | 256.62 | 282.57 | 305.11 | 189.55 | 324.51 | 271.67 | 52.44 |
| | IMAP | 165.29 | 197.35 | 134.76 | 180.43 | 193.78 | 174.32 | 25.46 |
| AdvDM(+) | baseline | 458.24 | 479.99 | 514.19 | 369.13 | 451.29 | 454.57 | 53.68 |
| | IMAP | 195.40 | 299.00 | 304.53 | 239.82 | 268.02 | 261.35 | 45.11 |
| AdvDM(-) | baseline | 138.44 | 134.26 | 119.14 | 146.03 | 128.38 | 133.25 | 10.17 |
| | IMAP | 154.25 | 189.95 | 99.28 | 163.63 | 139.92 | 149.41 | 33.44 |
| Glaze | baseline | 222.75 | 274.01 | 309.87 | 216.54 | 325.11 | 269.65 | 49.33 |
| | IMAP | 216.16 | 263.94 | 254.93 | 190.87 | 293.04 | 243.79 | 40.37 |
| SDS(+) | baseline | 358.61 | 491.20 | 490.71 | 354.68 | 453.02 | 429.64 | 68.43 |
| | IMAP | 210.95 | 297.22 | 136.98 | 172.61 | 326.30 | 228.81 | 80.77 |
| SDS(-) | baseline | 155.51 | 125.45 | 98.07 | 153.87 | 138.23 | 134.22 | 23.66 |
| | IMAP | 168.59 | 169.52 | 109.93 | 162.32 | 156.14 | 153.30 | 24.83 |
| SDST | baseline | 266.36 | 297.70 | 333.18 | 206.88 | 292.43 | 279.31 | 46.97 |
| | IMAP | 217.80 | 231.36 | 186.38 | 215.02 | 204.31 | 210.98 | 16.79 |
| *Diff prompt: $c_{diff}$* | | | | | | | | |
| AntiDB | baseline | 211.68 | 211.60 | 245.75 | 213.24 | 233.14 | 223.08 | 15.60 |
| | IMAP | 172.55 | 121.74 | 85.65 | 176.42 | 173.89 | 146.05 | 40.74 |
| AdvDM(+) | baseline | 324.12 | 420.31 | 418.16 | 301.56 | 323.50 | 357.53 | 57.06 |
| | IMAP | 183.27 | 152.40 | 121.18 | 141.27 | 203.48 | 160.32 | 32.97 |
| AdvDM(-) | baseline | 242.11 | 151.43 | 170.15 | 129.85 | 192.89 | 177.29 | 43.05 |
| | IMAP | 195.72 | 158.74 | 124.13 | 147.29 | 158.70 | 156.92 | 25.88 |
| Glaze | baseline | 207.73 | 232.04 | 220.53 | 173.44 | 196.92 | 206.13 | 22.55 |
| | IMAP | 193.24 | 141.10 | 117.04 | 188.56 | 163.52 | 160.69 | 32.15 |
| SDS(+) | baseline | 312.83 | 338.35 | 340.26 | 360.43 | 348.85 | 340.15 | 17.58 |
| | IMAP | 186.13 | 169.56 | 93.18 | 151.77 | 167.28 | 153.58 | 35.89 |
| SDS(-) | baseline | 225.75 | 141.89 | 144.13 | 125.69 | 188.91 | 165.27 | 41.16 |
| | IMAP | 197.82 | 136.66 | 118.46 | 156.35 | 166.38 | 155.13 | 30.15 |
| SDST | baseline | 338.47 | 282.64 | 299.53 | 241.14 | 295.53 | 291.46 | 35.01 |
| | IMAP | 208.51 | 154.60 | 147.27 | 166.32 | 159.35 | 167.21 | 24.11 |

Table 10: **Quantitative evaluation on CelebA-HQ under different sampling prompt settings (PSNR↑).**

| Adersarial Perturbations | Dataset: CelebA-HQ, Metric: PSNR↑ | | | | | | | |
|---|---|---|---|---|---|---|---|---|
| | Identity | 80 | 95 | 103 | 104 | 108 | mean | std |
| Both prompts: $c_{train}, c_{diff}$ | | | | | | | | |
| AntiDB | baseline | 10.09 | 10.39 | 10.79 | 10.34 | 9.83 | 10.29 | 0.36 |
| | IMAP | 10.95 | 10.79 | 12.42 | 10.49 | 10.75 | 11.08 | 0.76 |
| AdvDM(+) | baseline | 9.55 | 9.42 | 10.21 | 9.55 | 9.08 | 9.56 | 0.41 |
| | IMAP | 11.15 | 10.85 | 11.71 | 10.54 | 10.12 | 10.87 | 0.60 |
| AdvDM(-) | baseline | 11.08 | 12.15 | 12.60 | 11.68 | 11.22 | 11.75 | 0.63 |
| | IMAP | 11.59 | 12.09 | 13.27 | 11.55 | 11.97 | 12.09 | 0.70 |
| Glaze | baseline | 9.94 | 9.79 | 10.22 | 9.55 | 9.96 | 9.89 | 0.25 |
| | IMAP | 10.62 | 10.81 | 11.72 | 9.83 | 10.48 | 10.69 | 0.68 |
| SDS(+) | baseline | 9.54 | 10.18 | 10.32 | 9.19 | 9.66 | 9.78 | 0.47 |
| | IMAP | 10.63 | 10.50 | 12.09 | 10.98 | 10.53 | 10.95 | 0.67 |
| SDS(-) | baseline | 11.41 | 11.90 | 13.20 | 11.56 | 11.30 | 11.87 | 0.77 |
| | IMAP | 11.91 | 12.17 | 13.36 | 11.45 | 12.01 | 12.18 | 0.71 |
| SDST | baseline | 10.26 | 10.17 | 10.61 | 9.90 | 10.08 | 10.21 | 0.26 |
| | IMAP | 11.14 | 10.64 | 11.93 | 10.53 | 10.61 | 10.97 | 0.59 |
| Train prompt: $c_{train}$ | | | | | | | | |
| AntiDB | baseline | 10.21 | 10.33 | 10.85 | 10.44 | 9.86 | 10.34 | 0.36 |
| | IMAP | 10.29 | 9.68 | 11.15 | 9.61 | 10.36 | 10.22 | 0.62 |
| AdvDM(+) | baseline | 9.49 | 8.65 | 9.99 | 8.69 | 8.43 | 9.05 | 0.66 |
| | IMAP | 10.63 | 9.63 | 10.80 | 9.42 | 9.22 | 9.94 | 0.73 |
| AdvDM(-) | baseline | 11.58 | 11.26 | 13.11 | 11.45 | 11.40 | 11.76 | 0.76 |
| | IMAP | 11.35 | 11.34 | 13.28 | 10.63 | 11.58 | 11.64 | 0.99 |
| Glaze | baseline | 9.91 | 9.54 | 9.36 | 8.92 | 9.64 | 9.47 | 0.37 |
| | IMAP | 9.82 | 9.73 | 9.69 | 8.87 | 9.51 | 9.52 | 0.38 |
| SDS(+) | baseline | 9.74 | 9.11 | 9.94 | 8.93 | 9.06 | 9.36 | 0.45 |
| | IMAP | 9.44 | 9.61 | 11.01 | 10.14 | 9.39 | 9.92 | 0.68 |
| SDS(-) | baseline | 12.02 | 11.41 | 13.65 | 11.14 | 11.62 | 11.97 | 0.99 |
| | IMAP | 12.18 | 10.89 | 13.30 | 10.52 | 11.42 | 11.66 | 1.11 |
| SDST | baseline | 10.98 | 10.56 | 11.03 | 9.98 | 10.03 | 10.51 | 0.50 |
| | IMAP | 11.06 | 9.68 | 10.91 | 9.79 | 9.10 | 10.11 | 0.84 |
| Diff prompt: $c_{diff}$ | | | | | | | | |
| AntiDB | baseline | 9.96 | 10.44 | 10.73 | 10.25 | 9.79 | 10.23 | 0.37 |
| | IMAP | 11.62 | 11.91 | 13.68 | 11.38 | 11.15 | 11.95 | 1.01 |
| AdvDM(+) | baseline | 9.60 | 10.18 | 10.44 | 10.42 | 9.73 | 10.07 | 0.39 |
| | IMAP | 11.66 | 12.08 | 12.63 | 11.65 | 11.03 | 11.81 | 0.59 |
| AdvDM(-) | baseline | 10.58 | 13.03 | 12.09 | 11.91 | 11.05 | 11.73 | 0.95 |
| | IMAP | 11.83 | 12.85 | 13.25 | 12.47 | 12.37 | 12.55 | 0.54 |
| Glaze | baseline | 9.97 | 10.05 | 11.08 | 10.17 | 10.27 | 10.31 | 0.45 |
| | IMAP | 11.41 | 11.89 | 13.75 | 10.79 | 11.45 | 11.86 | 1.13 |
| SDS(+) | baseline | 9.34 | 11.25 | 10.69 | 9.45 | 10.26 | 10.20 | 0.81 |
| | IMAP | 11.82 | 11.40 | 13.16 | 11.81 | 11.67 | 11.97 | 0.69 |
| SDS(-) | baseline | 10.80 | 12.38 | 12.74 | 11.97 | 10.99 | 11.78 | 0.85 |
| | IMAP | 11.64 | 13.44 | 13.42 | 12.39 | 12.59 | 12.70 | 0.76 |
| SDST | baseline | 9.54 | 9.79 | 10.20 | 9.82 | 10.13 | 9.90 | 0.27 |
| | IMAP | 11.21 | 11.59 | 12.96 | 11.28 | 12.13 | 11.83 | 0.72 |

Table 11: **Quantitative evaluation on CelebA-HQ under different sampling prompt settings (SSIM↑).**

| Adersarial Perturbations | Dataset: CelebA-HQ, Metric: SSIM↑ | | | | | | | |
| --- | --- | --- | --- | --- | --- | --- | --- | --- |
| | Identity | 80 | 95 | 103 | 104 | 108 | mean | std |
| *Both prompts: $c_{train}, c_{diff}$* | | | | | | | | |
| AntiDB | baseline | 0.23 | 0.26 | 0.21 | 0.20 | 0.19 | 0.22 | 0.03 |
| | IMAP | 0.34 | 0.34 | 0.43 | 0.29 | 0.30 | 0.34 | 0.05 |
| AdvDM(+) | baseline | 0.20 | 0.21 | 0.22 | 0.21 | 0.19 | 0.21 | 0.01 |
| | IMAP | 0.34 | 0.36 | 0.36 | 0.31 | 0.31 | 0.34 | 0.02 |
| AdvDM(-) | baseline | 0.31 | 0.45 | 0.44 | 0.34 | 0.35 | 0.38 | 0.06 |
| | IMAP | 0.36 | 0.46 | 0.51 | 0.37 | 0.40 | 0.42 | 0.06 |
| Glaze | baseline | 0.20 | 0.21 | 0.20 | 0.20 | 0.21 | 0.20 | 0.01 |
| | IMAP | 0.27 | 0.31 | 0.34 | 0.25 | 0.27 | 0.29 | 0.04 |
| SDS(+) | baseline | 0.20 | 0.24 | 0.20 | 0.19 | 0.20 | 0.21 | 0.02 |
| | IMAP | 0.33 | 0.33 | 0.42 | 0.34 | 0.31 | 0.34 | 0.04 |
| SDS(-) | baseline | 0.34 | 0.45 | 0.48 | 0.34 | 0.34 | 0.39 | 0.07 |
| | IMAP | 0.37 | 0.47 | 0.50 | 0.37 | 0.39 | 0.42 | 0.06 |
| SDST | baseline | 0.22 | 0.29 | 0.27 | 0.23 | 0.24 | 0.25 | 0.03 |
| | IMAP | 0.33 | 0.41 | 0.42 | 0.33 | 0.36 | 0.37 | 0.04 |
| *Train prompt: $c_{train}$* | | | | | | | | |
| AntiDB | baseline | 0.19 | 0.21 | 0.18 | 0.20 | 0.16 | 0.19 | 0.02 |
| | IMAP | 0.27 | 0.22 | 0.31 | 0.20 | 0.24 | 0.25 | 0.05 |
| AdvDM(+) | baseline | 0.17 | 0.18 | 0.19 | 0.17 | 0.14 | 0.17 | 0.02 |
| | IMAP | 0.27 | 0.23 | 0.24 | 0.22 | 0.23 | 0.24 | 0.02 |
| AdvDM(-) | baseline | 0.27 | 0.38 | 0.42 | 0.28 | 0.32 | 0.33 | 0.07 |
| | IMAP | 0.31 | 0.40 | 0.48 | 0.31 | 0.36 | 0.37 | 0.07 |
| Glaze | baseline | 0.17 | 0.14 | 0.13 | 0.13 | 0.14 | 0.14 | 0.02 |
| | IMAP | 0.18 | 0.18 | 0.17 | 0.16 | 0.17 | 0.17 | 0.01 |
| SDS(+) | baseline | 0.19 | 0.19 | 0.18 | 0.18 | 0.17 | 0.18 | 0.01 |
| | IMAP | 0.24 | 0.23 | 0.33 | 0.28 | 0.21 | 0.26 | 0.05 |
| SDS(-) | baseline | 0.32 | 0.38 | 0.47 | 0.28 | 0.31 | 0.35 | 0.07 |
| | IMAP | 0.34 | 0.40 | 0.47 | 0.31 | 0.34 | 0.37 | 0.06 |
| SDST | baseline | 0.23 | 0.29 | 0.27 | 0.21 | 0.22 | 0.24 | 0.04 |
| | IMAP | 0.28 | 0.37 | 0.36 | 0.26 | 0.31 | 0.32 | 0.05 |
| *Diff prompt: $c_{diff}$* | | | | | | | | |
| AntiDB | baseline | 0.26 | 0.31 | 0.25 | 0.21 | 0.21 | 0.25 | 0.04 |
| | IMAP | 0.40 | 0.46 | 0.54 | 0.37 | 0.36 | 0.43 | 0.07 |
| AdvDM(+) | baseline | 0.23 | 0.25 | 0.25 | 0.25 | 0.23 | 0.24 | 0.01 |
| | IMAP | 0.40 | 0.48 | 0.49 | 0.41 | 0.39 | 0.43 | 0.04 |
| AdvDM(-) | baseline | 0.35 | 0.52 | 0.46 | 0.40 | 0.38 | 0.42 | 0.07 |
| | IMAP | 0.41 | 0.52 | 0.54 | 0.43 | 0.44 | 0.47 | 0.06 |
| Glaze | baseline | 0.23 | 0.28 | 0.26 | 0.27 | 0.28 | 0.26 | 0.02 |
| | IMAP | 0.36 | 0.43 | 0.51 | 0.34 | 0.36 | 0.40 | 0.07 |
| SDS(+) | baseline | 0.21 | 0.29 | 0.22 | 0.20 | 0.23 | 0.23 | 0.04 |
| | IMAP | 0.41 | 0.42 | 0.51 | 0.41 | 0.41 | 0.43 | 0.04 |
| SDS(-) | baseline | 0.36 | 0.51 | 0.50 | 0.40 | 0.38 | 0.43 | 0.07 |
| | IMAP | 0.40 | 0.54 | 0.53 | 0.43 | 0.44 | 0.47 | 0.06 |
| SDST | baseline | 0.21 | 0.30 | 0.27 | 0.26 | 0.26 | 0.26 | 0.03 |
| | IMAP | 0.38 | 0.45 | 0.48 | 0.40 | 0.42 | 0.43 | 0.04 |

Table 12: **Quantitative evaluation on CelebA-HQ under different sampling prompt settings (TOPIQ↑).**

| Adersarial Perturbations | Dataset: CelebA-HQ, Metric: TOPIQ↑ | | | | | | | |
|---|---|---|---|---|---|---|---|---|
| | Identity | 80 | 95 | 103 | 104 | 108 | mean | std |
| *Both prompts: $c_{train}, c_{diff}$* | | | | | | | | |
| AntiDB | baseline | 0.56 | 0.51 | 0.47 | 0.49 | 0.36 | 0.48 | 0.08 |
| | IMAP | 0.67 | 0.64 | 0.69 | 0.62 | 0.58 | 0.64 | 0.04 |
| AdvDM(+) | baseline | 0.17 | 0.09 | 0.04 | 0.17 | 0.13 | 0.12 | 0.06 |
| | IMAP | 0.59 | 0.47 | 0.49 | 0.57 | 0.51 | 0.53 | 0.05 |
| AdvDM(-) | baseline | 0.43 | 0.33 | 0.34 | 0.39 | 0.37 | 0.37 | 0.04 |
| | IMAP | 0.45 | 0.45 | 0.44 | 0.54 | 0.47 | 0.47 | 0.04 |
| Glaze | baseline | 0.52 | 0.42 | 0.40 | 0.47 | 0.52 | 0.47 | 0.06 |
| | IMAP | 0.58 | 0.54 | 0.50 | 0.53 | 0.54 | 0.54 | 0.03 |
| SDS(+) | baseline | 0.25 | 0.23 | 0.23 | 0.10 | 0.24 | 0.21 | 0.06 |
| | IMAP | 0.58 | 0.47 | 0.58 | 0.70 | 0.49 | 0.56 | 0.09 |
| SDS(-) | baseline | 0.36 | 0.34 | 0.32 | 0.36 | 0.36 | 0.35 | 0.02 |
| | IMAP | 0.48 | 0.43 | 0.38 | 0.49 | 0.42 | 0.44 | 0.04 |
| SDST | baseline | 0.43 | 0.35 | 0.46 | 0.39 | 0.35 | 0.40 | 0.05 |
| | IMAP | 0.55 | 0.36 | 0.43 | 0.43 | 0.55 | 0.47 | 0.08 |
| *Train prompt: $c_{train}$* | | | | | | | | |
| AntiDB | baseline | 0.42 | 0.38 | 0.44 | 0.42 | 0.27 | 0.39 | 0.07 |
| | IMAP | 0.51 | 0.48 | 0.69 | 0.47 | 0.41 | 0.51 | 0.10 |
| AdvDM(+) | baseline | 0.01 | 0.02 | 0.02 | 0.10 | 0.02 | 0.04 | 0.04 |
| | IMAP | 0.44 | 0.35 | 0.36 | 0.37 | 0.37 | 0.38 | 0.04 |
| AdvDM(-) | baseline | 0.39 | 0.37 | 0.40 | 0.37 | 0.37 | 0.38 | 0.02 |
| | IMAP | 0.37 | 0.40 | 0.42 | 0.45 | 0.38 | 0.40 | 0.03 |
| Glaze | baseline | 0.33 | 0.31 | 0.25 | 0.25 | 0.42 | 0.31 | 0.07 |
| | IMAP | 0.34 | 0.32 | 0.25 | 0.32 | 0.33 | 0.31 | 0.04 |
| SDS(+) | baseline | 0.18 | 0.13 | 0.08 | 0.11 | 0.12 | 0.13 | 0.04 |
| | IMAP | 0.40 | 0.28 | 0.50 | 0.62 | 0.29 | 0.42 | 0.14 |
| SDS(-) | baseline | 0.34 | 0.35 | 0.36 | 0.36 | 0.34 | 0.35 | 0.01 |
| | IMAP | 0.35 | 0.39 | 0.33 | 0.39 | 0.34 | 0.36 | 0.03 |
| SDST | baseline | 0.45 | 0.35 | 0.52 | 0.44 | 0.38 | 0.43 | 0.07 |
| | IMAP | 0.44 | 0.27 | 0.48 | 0.35 | 0.47 | 0.40 | 0.09 |
| *Diff prompt: $c_{diff}$* | | | | | | | | |
| AntiDB | baseline | 0.71 | 0.63 | 0.50 | 0.55 | 0.44 | 0.57 | 0.10 |
| | IMAP | 0.82 | 0.81 | 0.70 | 0.77 | 0.74 | 0.77 | 0.05 |
| AdvDM(+) | baseline | 0.33 | 0.16 | 0.05 | 0.25 | 0.24 | 0.21 | 0.11 |
| | IMAP | 0.74 | 0.59 | 0.62 | 0.77 | 0.64 | 0.67 | 0.08 |
| AdvDM(-) | baseline | 0.46 | 0.30 | 0.27 | 0.41 | 0.37 | 0.36 | 0.08 |
| | IMAP | 0.53 | 0.50 | 0.46 | 0.64 | 0.56 | 0.54 | 0.07 |
| Glaze | baseline | 0.70 | 0.54 | 0.55 | 0.68 | 0.63 | 0.62 | 0.07 |
| | IMAP | 0.82 | 0.77 | 0.74 | 0.74 | 0.75 | 0.76 | 0.03 |
| SDS(+) | baseline | 0.31 | 0.32 | 0.39 | 0.09 | 0.35 | 0.29 | 0.12 |
| | IMAP | 0.76 | 0.65 | 0.66 | 0.79 | 0.69 | 0.71 | 0.06 |
| SDS(-) | baseline | 0.38 | 0.33 | 0.28 | 0.36 | 0.37 | 0.35 | 0.04 |
| | IMAP | 0.60 | 0.47 | 0.43 | 0.58 | 0.50 | 0.52 | 0.07 |
| SDST | baseline | 0.42 | 0.35 | 0.40 | 0.34 | 0.31 | 0.36 | 0.04 |
| | IMAP | 0.65 | 0.46 | 0.37 | 0.52 | 0.63 | 0.53 | 0.12 |

Table 13: **Quantitative evaluation on CelebA-HQ under different sampling prompt settings (QAlign↑).**

| Adersarial Perturbations | Dataset: CelebA-HQ, Metric: QAlign↑ | | | | | | | |
|---|---|---|---|---|---|---|---|---|
| | Identity | 80 | 95 | 103 | 104 | 108 | mean | std |
| *Both prompts: $c_{train}, c_{diff}$* | | | | | | | | |
| AntiDB | baseline | 2.50 | 2.10 | 2.03 | 2.27 | 1.70 | 2.12 | 0.30 |
| | IMAP | 3.32 | 3.20 | 3.16 | 2.99 | 2.74 | 3.08 | 0.22 |
| AdvDM(+) | baseline | 1.74 | 1.37 | 1.35 | 1.52 | 1.34 | 1.46 | 0.17 |
| | IMAP | 2.76 | 2.17 | 2.38 | 2.64 | 2.63 | 2.51 | 0.24 |
| AdvDM(-) | baseline | 2.41 | 1.98 | 1.84 | 2.02 | 2.21 | 2.09 | 0.22 |
| | IMAP | 2.57 | 2.63 | 2.40 | 2.88 | 2.68 | 2.63 | 0.17 |
| Glaze | baseline | 2.74 | 1.89 | 1.94 | 2.32 | 2.24 | 2.23 | 0.34 |
| | IMAP | 3.26 | 2.71 | 2.60 | 2.78 | 2.58 | 2.78 | 0.28 |
| SDS(+) | baseline | 1.95 | 1.52 | 1.56 | 1.57 | 1.51 | 1.62 | 0.18 |
| | IMAP | 2.96 | 2.41 | 2.78 | 3.34 | 2.56 | 2.81 | 0.36 |
| SDS(-) | baseline | 2.13 | 1.97 | 1.85 | 1.95 | 2.12 | 2.01 | 0.12 |
| | IMAP | 2.72 | 2.41 | 2.15 | 2.59 | 2.45 | 2.46 | 0.21 |
| SDST | baseline | 2.85 | 2.32 | 3.13 | 2.24 | 2.38 | 2.58 | 0.39 |
| | IMAP | 2.85 | 1.96 | 2.25 | 2.31 | 2.72 | 2.42 | 0.36 |
| *Train prompt: $c_{train}$* | | | | | | | | |
| AntiDB | baseline | 1.98 | 1.67 | 1.89 | 2.02 | 1.61 | 1.84 | 0.19 |
| | IMAP | 2.39 | 2.29 | 2.71 | 2.08 | 1.89 | 2.27 | 0.31 |
| AdvDM(+) | baseline | 1.87 | 1.38 | 1.35 | 1.43 | 1.44 | 1.49 | 0.22 |
| | IMAP | 1.94 | 1.44 | 1.45 | 1.58 | 1.75 | 1.63 | 0.21 |
| AdvDM(-) | baseline | 2.22 | 2.05 | 2.09 | 1.98 | 2.22 | 2.11 | 0.11 |
| | IMAP | 2.25 | 2.33 | 2.29 | 2.40 | 2.25 | 2.30 | 0.06 |
| Glaze | baseline | 1.93 | 1.56 | 1.36 | 1.40 | 2.01 | 1.65 | 0.30 |
| | IMAP | 2.06 | 1.76 | 1.37 | 1.68 | 1.67 | 1.71 | 0.25 |
| SDS(+) | baseline | 1.86 | 1.56 | 1.45 | 1.54 | 1.39 | 1.56 | 0.18 |
| | IMAP | 2.02 | 1.41 | 2.24 | 2.76 | 1.55 | 2.00 | 0.55 |
| SDS(-) | baseline | 2.10 | 1.96 | 1.93 | 1.98 | 2.04 | 2.00 | 0.07 |
| | IMAP | 2.10 | 2.20 | 2.01 | 2.11 | 2.07 | 2.10 | 0.07 |
| SDST | baseline | 2.70 | 2.54 | 3.42 | 2.46 | 2.63 | 2.75 | 0.38 |
| | IMAP | 2.55 | 1.48 | 2.34 | 2.00 | 2.18 | 2.11 | 0.41 |
| *Diff prompt: $c_{diff}$* | | | | | | | | |
| AntiDB | baseline | 3.03 | 2.53 | 2.15 | 2.52 | 1.79 | 2.40 | 0.46 |
| | IMAP | 4.25 | 4.11 | 3.62 | 3.91 | 3.59 | 3.89 | 0.29 |
| AdvDM(+) | baseline | 1.60 | 1.35 | 1.35 | 1.60 | 1.25 | 1.43 | 0.16 |
| | IMAP | 3.57 | 2.90 | 3.30 | 3.69 | 3.51 | 3.39 | 0.31 |
| AdvDM(-) | baseline | 2.59 | 1.90 | 1.59 | 2.07 | 2.22 | 2.08 | 0.37 |
| | IMAP | 2.89 | 2.92 | 2.50 | 3.35 | 3.12 | 2.96 | 0.31 |
| Glaze | baseline | 3.54 | 2.21 | 2.52 | 3.25 | 2.46 | 2.80 | 0.57 |
| | IMAP | 4.46 | 3.66 | 3.83 | 3.88 | 3.49 | 3.86 | 0.37 |
| SDS(+) | baseline | 2.03 | 1.48 | 1.67 | 1.60 | 1.63 | 1.68 | 0.21 |
| | IMAP | 3.90 | 3.42 | 3.30 | 3.92 | 3.58 | 3.63 | 0.28 |
| SDS(-) | baseline | 2.16 | 1.98 | 1.77 | 1.93 | 2.20 | 2.01 | 0.18 |
| | IMAP | 3.33 | 2.63 | 2.29 | 3.07 | 2.83 | 2.83 | 0.40 |
| SDST | baseline | 3.00 | 2.10 | 2.83 | 2.01 | 2.13 | 2.41 | 0.47 |
| | IMAP | 3.16 | 2.45 | 2.16 | 2.63 | 3.28 | 2.73 | 0.47 |

## D.3 Supplementary for Section 5.3

We provide full evaluation results for comparative experiments in Tables 14 to 17. Specifically, the results in Tables 14 and 15 correspond to the comparative experiments under Noisy-Upscaling, and the results in Tables 16 and 17 correspond to the comparative experiments under IMPRESS.

Table 14: **Quantitative evaluation on VGGFace2 across multiple metrics under two experimental conditions: using only the purification method versus using both purification and IMAP.** We show results using Noisy-Upscaling as purification method in this table.

| Adversarial Perturbations | FID↓ | | PSNR↑ | | SSIM↑ | | TOPIQ↑ | | Qalign↑ | |
|---|---|---|---|---|---|---|---|---|---|---|
| | Purify only | Purify +IMAP | Purify only | Purify +IMAP | Purify only | Purify +IMAP | Purify only | Purify +IMAP | Purify only | Purify +IMAP |
| *Both prompts: $c_{train}, c_{diff}$* | | | | | | | | | | |
| AntiDB | **107.58** | 116.45 | 11.09 | **11.19** | 0.38 | **0.39** | 0.60 | **0.62** | 3.10 | **3.19** |
| AdvDM(+) | 204.58 | **143.68** | 9.95 | **10.80** | 0.25 | **0.34** | 0.38 | **0.55** | 1.75 | **2.71** |
| AdvDM(-) | **109.27** | 111.95 | 11.20 | **11.32** | 0.41 | **0.41** | 0.54 | **0.56** | 2.75 | **2.98** |
| Glaze | **109.29** | 112.54 | 11.11 | **11.36** | 0.37 | **0.39** | 0.56 | **0.59** | 2.84 | **3.02** |
| SDS(+) | 238.37 | **157.12** | 9.97 | **10.84** | 0.22 | **0.33** | 0.35 | **0.55** | 1.63 | **2.73** |
| SDS(-) | 114.54 | **111.15** | 10.91 | **11.51** | 0.39 | **0.42** | 0.51 | **0.53** | 2.63 | **2.78** |
| SDST | 170.38 | **146.44** | 10.27 | **10.85** | 0.31 | **0.37** | 0.46 | **0.52** | 2.38 | **2.75** |
| *Train prompt: $c_{train}$* | | | | | | | | | | |
| AntiDB | **120.89** | 140.75 | **10.61** | 10.03 | **0.33** | 0.31 | **0.52** | 0.51 | **2.77** | 2.63 |
| AdvDM(+) | 238.11 | **185.54** | **9.49** | 9.46 | 0.19 | **0.23** | 0.32 | **0.41** | 1.70 | **1.95** |
| AdvDM(-) | **125.60** | 138.33 | **10.52** | 10.03 | **0.35** | 0.34 | **0.52** | 0.49 | **2.74** | 2.61 |
| Glaze | **125.11** | 134.35 | **10.67** | 10.06 | **0.32** | 0.31 | **0.50** | 0.49 | **2.62** | 2.49 |
| SDS(+) | 287.61 | **213.41** | 9.44 | **9.54** | 0.16 | **0.22** | 0.28 | **0.40** | 1.54 | **1.96** |
| SDS(-) | **135.54** | 138.79 | **10.24** | 9.93 | **0.34** | 0.33 | **0.48** | 0.45 | **2.50** | 2.35 |
| SDST | **184.01** | 196.99 | **9.72** | 9.49 | 0.27 | **0.28** | **0.43** | 0.42 | **2.39** | 2.36 |
| *Diff prompt: $c_{diff}$* | | | | | | | | | | |
| AntiDB | 134.47 | **130.89** | 11.57 | **12.36** | 0.43 | **0.47** | 0.67 | **0.73** | 3.43 | **3.75** |
| AdvDM(+) | 211.49 | **143.27** | 10.41 | **12.14** | 0.32 | **0.45** | 0.44 | **0.70** | 1.81 | **3.47** |
| AdvDM(-) | 128.90 | **122.27** | 11.87 | **12.62** | 0.46 | **0.48** | 0.56 | **0.64** | 2.77 | **3.34** |
| Glaze | 132.21 | **129.67** | 11.56 | **12.65** | 0.42 | **0.47** | 0.62 | **0.69** | 3.07 | **3.54** |
| SDS(+) | 231.02 | **139.37** | 10.50 | **12.14** | 0.29 | **0.44** | 0.41 | **0.69** | 1.72 | **3.50** |
| SDS(-) | 136.67 | **122.26** | 11.58 | **13.09** | 0.44 | **0.50** | 0.54 | **0.60** | 2.75 | **3.21** |
| SDST | 193.27 | **140.02** | 10.82 | **12.20** | 0.35 | **0.45** | 0.49 | **0.62** | 2.38 | **3.14** |

Table 15: **Quantitative evaluation on CelebA-HQ across multiple metrics under two experimental conditions: using only the purification method versus using both purification and IMAP.** We show results using Noisy-Upscaling as purification method in this table.

| Adversarial Perturbations | FID↓ | | PSNR↑ | | SSIM↑ | | TOPIQ↑ | | Qalign↑ | |
|---|---|---|---|---|---|---|---|---|---|---|
| | Purify only | Purify +IMAP | Purify only | Purify +IMAP | Purify only | Purify +IMAP | Purify only | Purify +IMAP | Purify only | Purify +IMAP |
| *Both prompts: $c_{train}, c_{diff}$* | | | | | | | | | | |
| AntiDB | **104.75** | 107.37 | **12.25** | 11.98 | 0.38 | **0.39** | 0.62 | **0.64** | 3.04 | **3.14** |
| AdvDM(+) | 246.66 | **157.49** | 10.38 | **11.08** | 0.22 | **0.32** | 0.41 | **0.54** | 1.90 | **2.55** |
| AdvDM(-) | 117.21 | **114.89** | 11.60 | **11.93** | 0.40 | **0.42** | **0.65** | 0.64 | 3.21 | **3.23** |
| Glaze | **103.40** | 107.47 | **11.91** | 11.86 | 0.36 | **0.37** | **0.64** | 0.63 | **3.25** | 3.20 |
| SDS(+) | 234.57 | **152.93** | 10.40 | **11.14** | 0.21 | **0.32** | 0.37 | **0.55** | 1.75 | **2.61** |
| SDS(-) | 127.94 | **114.84** | 11.43 | **11.90** | 0.39 | **0.42** | **0.64** | 0.62 | 3.08 | **3.13** |
| SDST | 179.42 | **146.24** | 10.62 | **11.55** | 0.33 | **0.38** | 0.57 | **0.61** | 2.95 | **3.23** |
| *Train prompt: $c_{train}$* | | | | | | | | | | |
| AntiDB | **108.67** | 116.40 | **11.72** | 11.23 | 0.31 | **0.31** | **0.56** | 0.55 | **2.82** | 2.65 |
| AdvDM(+) | 281.44 | **203.55** | **10.32** | 10.10 | 0.17 | **0.21** | 0.34 | **0.37** | **1.83** | 1.75 |
| AdvDM(-) | 130.99 | **124.00** | **11.53** | 11.29 | 0.37 | **0.37** | **0.60** | 0.57 | **3.03** | 2.88 |
| Glaze | **114.06** | 120.80 | **11.42** | 11.11 | **0.28** | 0.28 | **0.52** | 0.50 | **2.74** | 2.57 |
| SDS(+) | 270.68 | **199.50** | 10.30 | **10.36** | 0.17 | **0.23** | 0.32 | **0.39** | 1.71 | **1.85** |
| SDS(-) | 140.76 | **123.63** | **11.50** | 11.33 | 0.36 | **0.37** | **0.60** | 0.56 | **2.92** | 2.82 |
| SDST | 203.01 | **195.77** | 10.56 | **10.68** | 0.30 | **0.31** | **0.52** | 0.51 | **2.88** | 2.80 |
| *Diff prompt: $c_{diff}$* | | | | | | | | | | |
| AntiDB | 135.93 | **132.39** | **12.79** | 12.73 | 0.45 | **0.47** | 0.67 | **0.73** | 3.26 | **3.62** |
| AdvDM(+) | 246.01 | **151.13** | 10.44 | **12.06** | 0.26 | **0.42** | 0.47 | **0.72** | 1.98 | **3.35** |
| AdvDM(-) | 143.89 | **140.14** | 11.68 | **12.56** | 0.43 | **0.47** | 0.69 | **0.71** | 3.40 | **3.57** |
| Glaze | **130.28** | 133.12 | 12.39 | **12.61** | 0.44 | **0.45** | 0.76 | **0.76** | 3.76 | **3.83** |
| SDS(+) | 234.92 | **149.10** | 10.50 | **11.92** | 0.25 | **0.41** | 0.42 | **0.71** | 1.79 | **3.36** |
| SDS(-) | 156.42 | **142.28** | 11.35 | **12.48** | 0.41 | **0.46** | 0.67 | **0.69** | 3.25 | **3.42** |
| SDST | 198.90 | **141.33** | 10.67 | **12.42** | 0.36 | **0.45** | 0.61 | **0.72** | 3.02 | **3.66** |

Table 16: **Quantitative evaluation on VGGFace2 across multiple metrics under two experimental conditions: using only the purification method versus using both purification and IMAP.** We show results using IMPRESS as purification method in this table.

| Adversarial Perturbations | FID↓ | | PSNR↑ | | SSIM↑ | | TOPIQ↑ | | Qalign↑ | |
|---|---|---|---|---|---|---|---|---|---|---|
| | Purify only | Purify +IMAP | Purify only | Purify +IMAP | Purify only | Purify +IMAP | Purify only | Purify +IMAP | Purify only | Purify +IMAP |
| *Both prompts: $c_{train}, c_{diff}$* | | | | | | | | | | |
| AntiDB | 312.22 | **167.46** | 9.70 | **11.12** | 0.21 | **0.35** | 0.28 | **0.55** | 1.55 | **2.69** |
| AdvDM(+) | 388.10 | **190.75** | 9.36 | **10.61** | 0.20 | **0.34** | 0.17 | **0.52** | 1.59 | **2.54** |
| AdvDM(-) | 145.07 | **137.96** | 10.87 | **11.58** | 0.40 | **0.43** | 0.31 | **0.39** | 1.76 | **2.19** |
| Glaze | 236.11 | **179.68** | 9.16 | **10.43** | 0.22 | **0.32** | 0.34 | **0.51** | 1.61 | **2.55** |
| SDS(+) | 350.50 | **198.51** | 9.46 | **10.71** | 0.20 | **0.33** | 0.22 | **0.51** | 1.52 | **2.60** |
| SDS(-) | **140.37** | 144.29 | 10.77 | **11.33** | 0.39 | **0.41** | 0.34 | **0.40** | 1.83 | **2.30** |
| SDST | 346.76 | **216.40** | 9.47 | **10.47** | 0.26 | **0.34** | 0.30 | **0.42** | 2.22 | **2.37** |
| *Train prompt: $c_{train}$* | | | | | | | | | | |
| AntiDB | 391.80 | **232.23** | 8.94 | **9.67** | 0.16 | **0.24** | 0.17 | **0.44** | 1.45 | **1.94** |
| AdvDM(+) | 450.99 | **260.78** | 8.87 | **9.12** | 0.17 | **0.24** | 0.06 | **0.39** | 1.57 | **1.77** |
| AdvDM(-) | **162.73** | 170.47 | **10.62** | 10.26 | **0.36** | 0.36 | 0.34 | 0.34 | 1.82 | **1.91** |
| Glaze | 250.48 | **233.78** | 8.57 | **8.72** | 0.17 | **0.20** | 0.29 | **0.34** | 1.30 | **1.59** |
| SDS(+) | 418.29 | **292.22** | 8.92 | **9.37** | 0.16 | **0.22** | 0.12 | **0.36** | 1.40 | **1.73** |
| SDS(-) | **159.93** | 186.33 | **10.45** | 10.30 | 0.35 | **0.35** | **0.35** | 0.33 | 1.82 | **1.95** |
| SDST | 347.88 | **291.30** | **9.35** | 9.33 | 0.24 | **0.25** | **0.35** | 0.34 | **2.49** | 2.13 |
| *Diff prompt: $c_{diff}$* | | | | | | | | | | |
| AntiDB | 284.70 | **154.43** | 10.45 | **12.57** | 0.26 | **0.47** | 0.39 | **0.67** | 1.64 | **3.45** |
| AdvDM(+) | 373.00 | **171.92** | 9.86 | **12.10** | 0.24 | **0.43** | 0.27 | **0.65** | 1.61 | **3.31** |
| AdvDM(-) | 165.92 | **144.94** | 11.12 | **12.90** | 0.43 | **0.49** | 0.29 | **0.44** | 1.69 | **2.46** |
| Glaze | 269.09 | **174.24** | 9.75 | **12.14** | 0.27 | **0.44** | 0.40 | **0.69** | 1.92 | **3.52** |
| SDS(+) | 326.66 | **159.02** | 10.00 | **12.05** | 0.23 | **0.44** | 0.32 | **0.66** | 1.64 | **3.48** |
| SDS(-) | 160.05 | **141.42** | 11.09 | **12.36** | 0.43 | **0.47** | 0.33 | **0.48** | 1.85 | **2.65** |
| SDST | 391.35 | **188.50** | 9.59 | **11.61** | 0.28 | **0.42** | 0.25 | **0.49** | 1.95 | **2.61** |

Table 17: **Quantitative evaluation on CelebA-HQ across multiple metrics under two experimental conditions: using only the purification method versus using both purification and IMAP.** We show results using IMPRESS as purification method in this table.

| Adversarial Perturbations | FID↓ | | PSNR↑ | | SSIM↑ | | TOPIQ↑ | | Qalign↑ | |
|---|---|---|---|---|---|---|---|---|---|---|
| | Purify only | Purify +IMAP | Purify only | Purify +IMAP | Purify only | Purify +IMAP | Purify only | Purify +IMAP | Purify only | Purify +IMAP |
| *Both prompts: $c_{train}, c_{diff}$* | | | | | | | | | | |
| AntiDB | 220.30 | **145.49** | 10.54 | **11.18** | 0.24 | **0.33** | 0.54 | **0.64** | 2.42 | **3.08** |
| AdvDM(+) | 373.32 | **172.21** | 9.50 | **10.77** | 0.18 | **0.33** | 0.15 | **0.55** | 1.59 | **2.61** |
| AdvDM(-) | 137.72 | **133.00** | 11.30 | **12.06** | 0.40 | **0.42** | 0.34 | **0.39** | 1.89 | **2.12** |
| Glaze | 205.91 | **170.79** | 9.95 | **10.75** | 0.23 | **0.32** | 0.47 | **0.56** | 2.26 | **2.72** |
| SDS(+) | 363.77 | **172.74** | 9.81 | **11.13** | 0.19 | **0.34** | 0.16 | **0.53** | 1.63 | **2.70** |
| SDS(-) | **139.57** | 141.03 | 11.34 | **12.09** | 0.40 | **0.42** | 0.32 | **0.37** | 1.87 | **2.10** |
| SDST | 256.47 | **181.84** | 9.98 | **10.98** | 0.27 | **0.36** | 0.36 | **0.41** | **2.53** | 2.34 |
| *Train prompt: $c_{train}$* | | | | | | | | | | |
| AntiDB | 265.08 | **192.51** | 10.25 | **10.39** | 0.19 | **0.25** | 0.41 | **0.52** | 2.13 | **2.22** |
| AdvDM(+) | 441.59 | **240.74** | 9.20 | **9.78** | 0.15 | **0.24** | 0.04 | **0.43** | 1.58 | **1.83** |
| AdvDM(-) | **133.97** | 138.39 | 11.39 | **11.44** | **0.38** | 0.37 | 0.37 | **0.38** | **2.08** | 2.06 |
| Glaze | 227.72 | 219.94 | **9.94** | 9.65 | 0.20 | **0.23** | **0.40** | 0.39 | **2.15** | 1.90 |
| SDS(+) | 430.13 | **237.90** | 9.49 | **10.17** | 0.16 | **0.25** | 0.07 | **0.39** | 1.61 | **1.95** |
| SDS(-) | **142.32** | 156.05 | 11.52 | **11.61** | **0.39** | 0.37 | 0.35 | **0.35** | 2.00 | **2.01** |
| SDST | 280.36 | **237.88** | **10.70** | 10.14 | 0.27 | **0.30** | **0.36** | 0.34 | **2.69** | 2.13 |
| *Diff prompt: $c_{diff}$* | | | | | | | | | | |
| AntiDB | 216.89 | **145.99** | 10.83 | **11.98** | 0.29 | **0.42** | 0.67 | **0.77** | 2.71 | **3.94** |
| AdvDM(+) | 345.51 | **163.59** | 9.81 | **11.76** | 0.21 | **0.42** | 0.25 | **0.66** | 1.61 | **3.41** |
| AdvDM(-) | 180.27 | **164.18** | 11.21 | **12.69** | 0.42 | **0.47** | 0.30 | **0.39** | 1.70 | **2.19** |
| Glaze | 224.49 | **174.78** | 9.96 | **11.85** | 0.26 | **0.41** | 0.53 | **0.72** | 2.38 | **3.55** |
| SDS(+) | 339.61 | **158.87** | 10.14 | **12.09** | 0.22 | **0.43** | 0.25 | **0.68** | 1.64 | **3.45** |
| SDS(-) | 172.67 | **164.75** | 11.17 | **12.57** | 0.42 | **0.46** | 0.29 | **0.39** | 1.75 | **2.18** |
| SDST | 264.82 | **177.65** | 9.26 | **11.82** | 0.28 | **0.42** | 0.36 | **0.48** | 2.37 | **2.55** |

## D.4 EVALUATING IMAP TRANSFERABILITY WITH FIXED-MASKS ACROSS DATASETS

We analyzed kernel masking patterns across all experiments and found that, under the same perturbation, certain kernels were frequently masked across different identities and even across datasets. Under the same perturbation, specific kernels tended to be frequently masked across different identities and even across different datasets. In contrast, different perturbations could lead to significant local variations in the masking patterns. Notably, SDST resulted in more dispersed masking compared to AntiDB, as illustrated in the Figure 10. This suggests that critical kernels are more dependent on the type of perturbation than on the dataset. To verify this, we constructed a perturbation-specific fixed mask based on VGGFace2 statistics and applied it to both VGGFace2 and CelebA-HQ. Results in Table 18 support this hypothesis.

Table 18: **Quantitative evaluation on different datasets using both sampling prompts under the fixed-mask setting.** The fixed mask is derived from prior VGGFace2 experiments by selecting kernels that were masked in at least 4 out of 5 times for each perturbation, and applied to both datasets to evaluate effectiveness and transferability.

| Adversarial Perturbations | FID↓ | | PSNR↑ | | SSIM↑ | | TOPIQ↑ | | QAlign↑ | |
|---|---|---|---|---|---|---|---|---|---|---|
| | Baseline | IMAP | Baseline | IMAP | Baseline | IMAP | Baseline | IMAP | Baseline | IMAP |
| Both prompts under VGGFace2 | | | | | | | | | | |
| AntiDB | 377.84 | **162.43** | 9.38 | **11.00** | 0.21 | **0.35** | 0.20 | **0.55** | 1.55 | **2.72** |
| AdvDM(+) | 397.33 | **194.55** | 9.05 | **10.79** | 0.20 | **0.34** | 0.13 | **0.53** | 1.43 | **2.45** |
| AdvDM(-) | 147.99 | **136.04** | 11.12 | **11.77** | 0.38 | **0.43** | 0.34 | **0.45** | 1.97 | **2.56** |
| Glaze | 224.52 | **170.16** | 9.11 | **10.43** | 0.21 | **0.31** | 0.34 | **0.51** | 1.60 | **2.59** |
| SDS(+) | 378.86 | **190.29** | 9.38 | **10.80** | 0.21 | **0.34** | 0.16 | **0.54** | 1.49 | **2.78** |
| SDS(-) | 147.67 | **143.03** | 11.10 | **11.77** | 0.38 | **0.42** | 0.32 | **0.40** | 1.91 | **2.33** |
| SDST | 327.41 | **202.82** | 10.02 | **10.98** | 0.28 | **0.36** | 0.35 | **0.47** | 2.12 | **2.60** |
| Both prompts under CelebA-HQ | | | | | | | | | | |
| AntiDB | 226.25 | **135.11** | 10.29 | **11.05** | 0.22 | **0.33** | 0.48 | **0.64** | 2.12 | **3.11** |
| AdvDM(+) | 387.31 | **184.19** | 9.56 | **10.88** | 0.21 | **0.33** | 0.12 | **0.52** | 1.46 | **2.51** |
| AdvDM(-) | 137.77 | **131.34** | 11.75 | **12.14** | 0.38 | **0.42** | 0.37 | **0.47** | 2.09 | **2.61** |
| Glaze | 215.03 | **167.13** | 9.89 | **10.76** | 0.20 | **0.29** | 0.47 | **0.54** | 2.23 | **2.79** |
| SDS(+) | 367.32 | **164.27** | 9.78 | **11.01** | 0.21 | **0.35** | 0.21 | **0.56** | 1.62 | **2.80** |
| SDS(-) | **132.55** | 133.94 | 11.87 | **12.14** | 0.39 | **0.42** | 0.35 | **0.44** | 2.01 | **2.48** |
| SDST | 268.10 | **169.12** | 10.21 | **11.08** | 0.25 | **0.37** | 0.40 | **0.47** | **2.58** | 2.51 |

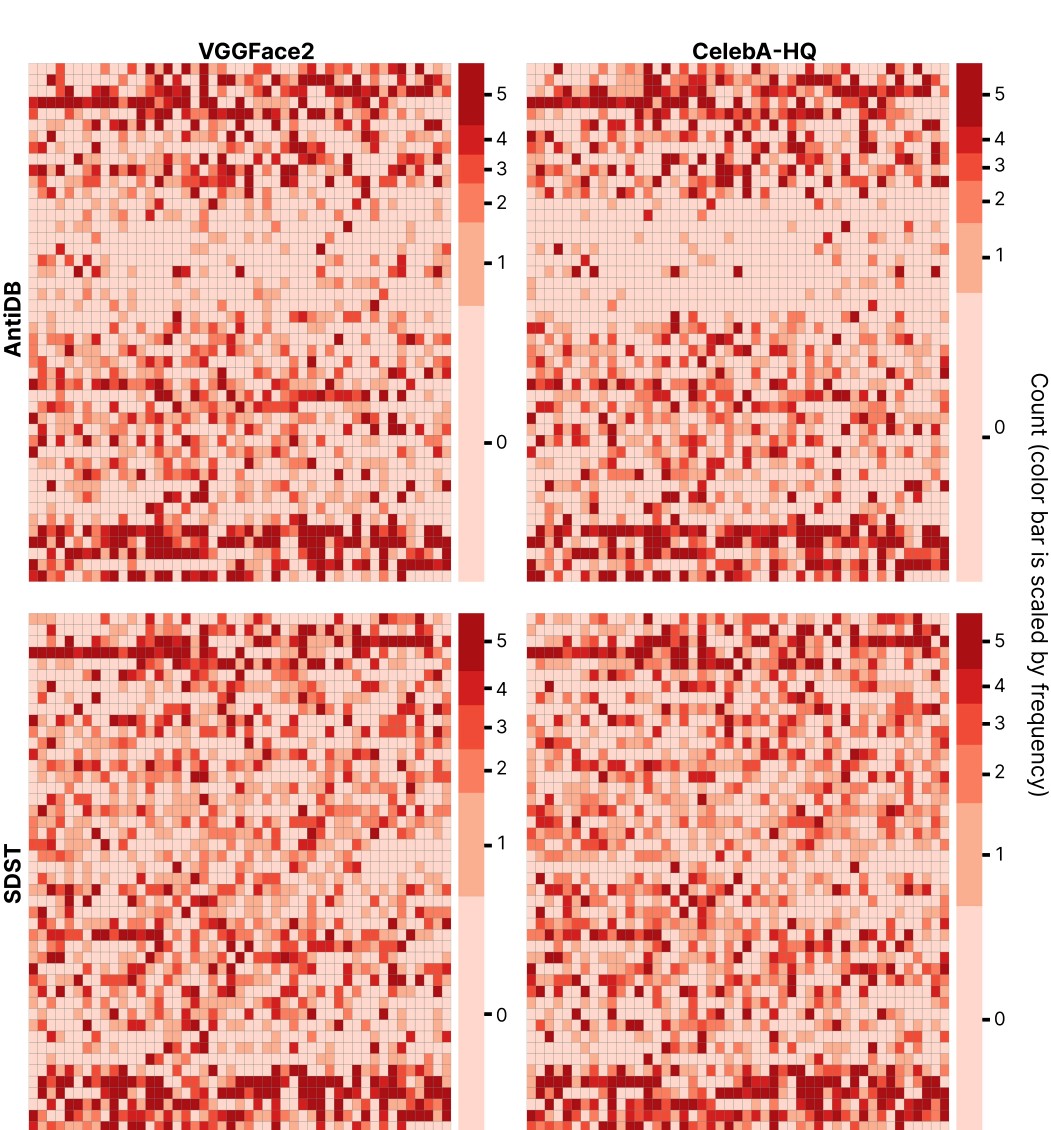

Figure 10: **Counts of masked times for each kernel under different perturbations (AntiDB and SDST) and datasets.** We consider all the kernels that have been masked at least once in all experiments, the color of each square denotes masked times of each kernel under same datasets.

