# OpenReview forum: "Inversion-Guided Weight Masking and Patching: Purification-Free Defense for Diffusion Models against Adversarial Perturbation"
_ICLR.cc/2026/Conference — ICLR 2026 Conference Withdrawn Submission_

### Official Review · Reviewer_1Pya · 2025-10-21

**Soundness:** 2
**Presentation:** 2
**Contribution:** 2
**Rating:** 4
**Confidence:** 3

**Summary:**

When fine-tuning diffusion models, the pre-trained model may be vulnerable to adversarial examples. Previous methods tend to first purify the training data, and then fine-tune the model. This work proposes another strategy. Motivated by the phenomenon of shortcut learning, the authors investigate the generation process via DDIM inversion. The perturbation-sensitive activations are masked, while the concept-relevant activations are preserved.

**Strengths:**

The authors have conducted experiments on different attacks. The proposed method can also be combined with existing purification tasks.

**Weaknesses:**

1. When fine-tuning, the authors assume all data are perturbed, and use the prompt `a photo of noised sks`. If we do not know whether the sample is adversarial, how could we choose the prompt? If one example is clean but the prompt is `noised sks`, the fine-tuned model may generate wrong reference datasets, which may influence all subsequent steps.

2. Some steps are heuristic. Why should we define the distance in Eq. 7? Why does $S_{sim}$ capture perturbation-induced shortcuts? At first glance, the similar activations may correspond to shared semantic features, such as `sks` and the class.

3. There is no ablation study about the proposed components, such as prompt remapping, masking, and patching.

4. It seems that the authors only use one text prompt (`a photo of xxx`). Can the model generate the subject or style with more complex prompts such as in DreamBooth?

5. Would the proposed method increase inference time?

**Questions:**

Please see Weaknesses

---

### Official Review · Reviewer_rKF3 · 2025-10-29

**Soundness:** 3
**Presentation:** 1
**Contribution:** 2
**Rating:** 4
**Confidence:** 4

**Summary:**

This paper proposes ​IMAP, a defense method that protects customized diffusion models from the detrimental effects of ​adversarial perturbations​ without requiring data purification. The core idea of IMAP is to repair the model itself after it has been customized on perturbed data, rather than cleaning the data beforehand. The proposed IMAP method is evaluated through extensive experiments on the CelebA-HQ and VGGFace2 datasets under various adversarial perturbations

**Strengths:**

This approach demonstrates a certain degree of innovation by repairing the weights of diffusion models to achieve adversarial defense.

**Weaknesses:**

1. The presentation of images and the quality of writing are relatively poor, making the content difficult to understand.
2. The paper employs a relatively large number of hyperparameters, which poses significant challenges for practical optimization.
3. The experimental section has notable weaknesses, primarily the lack of a comprehensive comparison with data purification methods.

**Questions:**

1. In the experimental section, the reported PSNR value is only 10 dB, which is exceptionally low. Could the authors provide a justification for this result?
2. Regarding the weight masking strategy, the paper appears to lack an ablation study to determine the optimal proportion of weights to mask. The chosen threshold does not seem to be rigorously established.

---

### Official Review · Reviewer_R2gM · 2025-10-29

**Soundness:** 3
**Presentation:** 2
**Contribution:** 3
**Rating:** 2
**Confidence:** 3

**Summary:**

This paper addresses the vulnerability of customised diffusion models to adversarial perturbations in training data. Unlike existing purification-based defences, which remove perturbations from the data prior to training, the authors propose IMAP: a purification-free method that repairs models at the parameter level following training on perturbed data. IMAP operates through three stages: fine-tuning, weight masking, and patching. Experiments demonstrate improvements in generation quality.

**Strengths:**

1. The paper offers a fresh perspective. Rather than relying on data purification, it re-examines the issue of defending diffusion models from the perspective of model restoration. The key insight is that adversarial perturbations exploit vulnerabilities in the model's internal components arising from shortcut learning. The author then proposes that these vulnerabilities can be rectified through weight masking. This provides a valuable alternative to existing purification-based approaches.

2. IMAP offers advantages in terms of efficiency and applicability. Its computational cost is independent of the size of the training dataset, requiring only a fixed amount of overhead, which makes it more scalable than purification methods. Furthermore, the method can operate without access to clean reference data, broadening the applicability of diffusion model defence to scenarios where clean samples are unavailable or difficult to obtain. Experimental results also demonstrate that IMAP can be combined with purification techniques to enhance robustness.

**Weaknesses:**

1. The method is complex, involving numerous design choices and hyperparameters, which makes it difficult to evaluate fairly. There are three stages, each with different prompt configurations. The weight-masking pipeline is complicated and incorporates both PCA-based feature reconstruction and Wasserstein distance. As the exposition is brief, the pipeline is difficult to reproduce. A clearer rationale and more detailed implementation would be helpful. More critically, the paper lacks ablation and sensitivity analyses to isolate the contribution of each component and assess robustness to hyperparameter variation.

2. Several key design decisions appear ad hoc and are based on empirical observations rather than a principled analysis. For instance, prompt remapping using 'noised sks' is introduced without clear motivation or evaluation demonstrating its necessity. Similarly, the emphasis on UpBlock1 and DownBlock2/3 in the U-Net is primarily justified by visual inspection of a small set of images. A more systematic study is required to determine whether these choices can be generalised or if they are dataset-specific heuristics.

3. The experimental validation is narrow in scope, raising questions about generalisability. Evaluations are limited to Stable Diffusion v2.1 with DreamBooth personalisation on facial images, with small sample sizes (20 images per experiment), and there are no direct comparisons with other defence methods. This makes it difficult to judge the method’s broader applicability, scalability and relative performance.

**Questions:**

1. In real scenarios, we may not always know whether an input or fine-tuning dataset has been adversarially perturbed.
How does IMAP behave when applied to legitimate, uncorrupted models?

---

### Official Review · Reviewer_nizj · 2025-10-31

**Soundness:** 2
**Presentation:** 2
**Contribution:** 2
**Rating:** 4
**Confidence:** 4

**Summary:**

This paper proposed a robust fine-tuning over diffusion model to make it resilient to adversarial perturbed training data. Based on the insights that diffusion models are fooled due to short-cut learning, they develope a framework based on fine-tuning, masking and patching: (1) fine-tune the model with perturbed data using new prompt (2) mask out kernels high related to adversarial effects (3) re-fine-tune the model with synthetic data. Finally, the authors did extensive experiments to show the effectiveness of the proposed method.

**Strengths:**

- This paper propose a complex robust fine-tuning pipeline to bypass adversarial perturbations in training images.
- The experiments are intensive and the quantitative results look better than baseline methods.

**Weaknesses:**

Methods:
- The proposed method is too complex and also it seems much slower.
- The authors only consider 'a photo of sks', there are not tests on other prompts 'a sks runnning' where dreambooth is designed to do so. It is unclear whether the proposed framework can work under other prompts, since they did fine-tuning on prompts with certrain formats, making it restricted.

Experiments:
- Only 16/255 is used as attack budget, smaller budget is not tested. It matters because the proposed framework may be sensitive to perturbation strength. Also, when the budget is small, baseline methods may be even better than proposed pipeline.
- Some other puritication methods e.g. GrIDPure [a] and PDM-Pure [c] are not cited or tested in the baseline. Also Mist-v2 [c] is not tested, which is SOTA attack against DreamBooth.
- In Figure 4, it seems that the purified fine-tuning results of IMAP look blury and still have artifacts (especially the last row).
- Why not comparing e.g. Noisy-Upscaling with IMAP? instead the authors compare NU with NU+IMAP.

Writing:
- The methods part of this paper is hard to read. The equations look unprofessional and hard to read, for equation please use \text{ } when there are texts inside.
- The proposed pipeline in Figure 1 does not reflect what they do in this paper, it is not as simple as recovering the features.

[a] Can Protective Perturbation Safeguard Personal Data from Being Exploited by Stable Diffusion?
[b] Pixel is a Barrier: Diffusion Models Are More Adversarially Robust Than We Think
[c] Targeted Attack Improves Protection against Unauthorized Diffusion Customization

**Questions:**

- In line:203  `we study a challenging scene ...`, in which setting can we acquire clean image during fine-tuning, what does that mean?

---

### Note · Authors · 2025-11-29

I have read and agree with the venue's withdrawal policy on behalf of myself and my co-authors.